# TWO-SHOT LEARNING OF CONTINUOUS INTERPOLATION USING A CONCEPTOR-AIDED RECURRENT AUTOENCODER

## ABSTRACT

Generalizing from only two time series towards unseen intermediate patterns poses a significant challenge in representation learning. In this paper, we introduce a novel representation learning algorithm, "Conceptor-Aided Recurrent Autoencoder" (CARAE), which leverages a conceptor-based regularization to learn to generate a continuous spectrum of intermediate temporal patterns while just being trained on two distinct examples. Here, conceptors, a linear subspace characterization of neuron activations, are employed to impose a low-dimensional geometrical bottleneck on the neural dynamics. During training, CARAE assembles a continuous and stable manifold between the two trained temporal patterns. Exploiting this manifold in the inference, CARAE facilitates continuous and phase-aligned interpolation between temporal patterns that are not linked within the training data. We demonstrate the effectiveness of the CARAE framework through comprehensive experiments on temporal pattern generation tasks and the generation of novel complex motion patterns based on the MoCap data set.

## 1 INTRODUCTION

Recurrent neural networks (RNNs) can capture complex temporal dependencies, making them an attractive choice for storing, retrieving, and generating temporal patterns (Salehinejad et al., 2017; Graves, 2013). As they process time series online, RNNs can avoid extensive buffering and hence computational overhead (Orvieto et al., 2023). However, despite a few examples in the domain of motor control for reinforcement learning scenarios (Merel et al., 2019b) and sketch drawing tasks (Ha & Eck, 2017), representational learning in RNNs is less advanced than in their static counterparts. Current RNN-based autoencoders struggle to capture long-range temporal concepts (Higgins et al., 2016; Girin et al., 2020; Heess et al., 2017), especially when used in the few-shot regime (Tran & Panangadan, 2022; Iwata & Kumagai, 2020). In that case, the challenge is to extract meaningful features from sparse training data and to enable interpolation between different temporal patterns, even those not encountered during training. Here, a critical point arises – formulating a meaningful bottleneck for the compression of spatiotemporal patterns.

One natural challenge of dynamical representational learning that is also tackled by roboticists and video game designers is the modeling of locomotion modeling (Song et al., 2021). For instance, when given training data of different behaviors, such as walking and running, the challenge is to compress to allow for meaningful interpolation (Chien & Wang, 2019). In motion modeling, one desires visually appealing intermediaries, which is difficult due to various instabilities like falling back to a fixed pose (Wang et al., 2017). In robotics and reinforcement learning, learning new skills may be massively accelerated by re-using previous skills based on their interpolation (Merel et al., 2019a).

Interestingly, despite the complexity and high dimensionality of locomotion patterns, they exhibit strong temporal signatures like high-level simplicity and approximate periodicity. This observation has spurred recent investigations into leveraging this temporal regularity. Traditional deep reinforcement learning (deepRL) methods often overlook temporal patterns, relying heavily on static Multi-Layer Perceptrons (MLPs). However, recent advancements have demonstrated the benefits of incorporating temporal priors to accelerate reinforcement learning of periodic-like behavior (Saanum

et al., 2023). Moreover, dynamical Variational Autoencoders (VAEs) have been designed to impose temporal priors, emphasizing the simplicity of high-level actions in representation learning, as proposed by Merel et al. (2019a). However, a gap remains in effectively introducing periodic inductive biases into neural networks at the network level (Belcák & Wattenhofer, 2022).

Here, our research question arises: *How to train a RNN to generate a continuum of temporal patterns in a two-shot manner, e.g. by learning the motion patterns of walking and running and then generating a continuous spectrum of intermediary patterns?* We will address the limitations of current RNN-based autoencoders by leveraging insights from a dynamical systems perspective. We specifically tackle the challenge of few-shot learning scenarios, where only a limited set of examples is used during training. While the task of interpolating between walking and running has been tackled in the deepRL setting to speed up learning (Merel et al., 2019b), we aim to develop and use new criteria for judging the quality of the learned representations (Bengio et al., 2014).

In the following, we draw inspiration from recent findings in computational neuroscience which indicate that low-dimensional dynamics play a key role in the brain. Exploiting computation in low-dimensional subspaces yielded successful models for various cognitive processes (Vyas et al., 2020). Dynamics that evolve on such low-dimensional subspaces can be also found within engineered RNNs (Carroll, 2021) and are exploited in the framework of reservoir computing (RC). Notably, recent RC methods generated advances in the inter-/extrapolation and abstraction of temporal patterns (Kim et al., 2021; Smith et al., 2022; Kong et al., 2023; Klos et al., 2020; Wyffels et al., 2014). Hereby, Kim et al. (2021) propose a method where during inference generalization towards close-by dynamics is achieved by training on several examples of a parametric pattern family.

Furthermore, (Jaeger, 2014) introduces a sophisticated control of low dimensional dynamics within high dimensional RNNs using the conceptor framework. Conceptors harnesses low-dimensional subspaces of RNNs to enable top-down control and dynamical mode switching through soft-projection matrices. Interestingly, while this work was designed within the framework of RC, its flexibility led to pioneering work in continual learning where it was combined with backpropagation to overcome catastrophic forgetting (He & Jaeger, 2018). Our research builds upon the insights derived from the conceptor-based characterization of low-dimensional dynamics, offering unique opportunities for designing bottlenecks in RNN-based autoencoders.

Our contributions are as follows. In section **??**, we provide a conceptual analysis grounded in examples from the literature to highlight the challenge of generating stable intermediate representations within RNNs. In section 3, we introduce our conceptor-based regularization yielding the CARAE framework and provide a geometric interpretation of the low dimensional bottleneck that is enforced on the RNN's neural dynamics. In section 4, we demonstrate the acquired capabilities on a toy task before we present continuous two-shot interpolation of motion patterns based on the MoCap motion modeling dataset. In section 4.3, we benchmark the quality of the learned representation by showing they can be used for robust feedback control.

## 2 OBJECTIVE AND ASSOCIATED CHALLENGES: FEW-SHOT TEMPORAL INTERPOLATION WITH AN AUTONOMOUS DYNAMICAL SYSTEM

The overall goal of our work is to train temporal pattern generators that are able to learn from a minimal number of temporal patterns and can then autonomously generate a continuous spectrum of intermediate patterns. Accordingly, we will focus on discrete-time dynamical systems such as RNNs that process data online by summarizing the past into a state. They can be generically described as input-driven dynamical systems:

$$x(n + 1) = F(x(n), u(n + 1); \boldsymbol{\theta}) \qquad \qquad y(n) = W_{out}x(n) \qquad (1)$$

where $F(\dots)$ is an element-wise applied nonlinear function, $x(n) \in \mathbb{R}^N$ is the systems state, $u(n) \in \mathbb{R}^M$ is the input signal, and $\boldsymbol{\theta}$ is a set of hyperparameters. Such RNNs can learn to retrieve temporal patterns in a supervised way by employing various learning frameworks such as reservoir computing (RC), real-time recurrent learning (RTRL), and backpropagation through time (BPTT).

In addition to this *input-driven mode*, the RNN can also be run in *autoregressive mode* where the RNN is decoupled from the external input signal and instead uses its own output:

$$x(n + 1) = F(x(n), y(n); \boldsymbol{\theta}) \qquad \qquad y(n) = W_{out}x(n) \qquad (2)$$

This is similar to large language models where the output is fed back as input to the model. However, the ability of RNNs to generalize to temporal patterns that were not present in the training data is limited, especially in this autoregressive mode, and until now can only be overcome with excessive amounts of training data. Hereby, we define four catastrophic scenarios that lead to an unwanted breakdown of the system dynamics.

- **Exploding dynamics:** Within the autoregressive mode, stability of the dynamics is not guaranteed and slight divergence from trained patterns can lead to continuously exploding prediction (Lukoševičius et al., 2012). This kind of exploding dynamics led to the development of the FORCE learning framework (Sussillo & Abbott, 2009) and noise-based regularization mechanism (Estébanez et al., 2019).

- **Interferences**: In the context of learning, while being input-driven, it is possible for state trajectories to intersect or cross each other. However, in the autonomous mode[1], this crossing leads to instabilities, as indicated in previous studies (Jaeger, 2014; Lu & Bassett, 2020).

- **Side dynamics:** Training a RNN on a few distinct temporal pattern yields a strong attraction towards these patterns. Initializing the RNN with intermediary examples and setting it into the autoregressive mode yields a sudden fall back to the learned patterns as shown in Fig. 1. Accordingly, intermediary dynamics are not supported by the RNN and these side dynamics can often only be overcome with more training data within the intermediary regime(Wyffels et al., 2014).

- **Fixed point dynamics:** Especially when employing control on RNNs to enforce intermediary states this can yield instabilities that cause the dynamics to collapse towards a fixed point (Jaeger, 2014). Once converged into the fixed point, periodic dynamics cannot be recovered as shown in Fig. 3 e) and g).

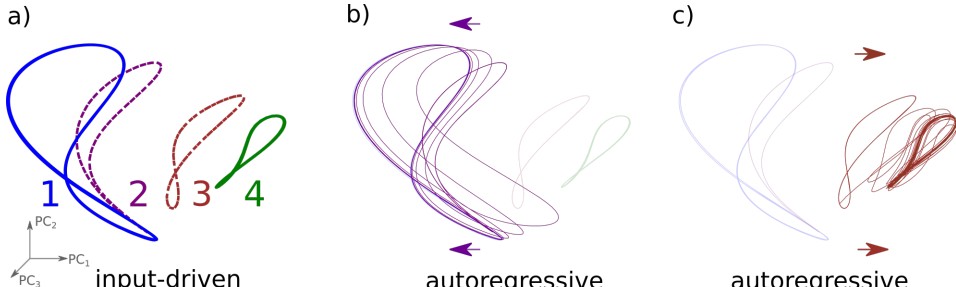

Figure 1: Illustration of the three largest principal components of the state space reverberations, $x(n)$, of the RNN trained with BPTT on sine wave pattern generation. a) Input driven state space for four different frequencies of the input sine waves. The RNN was trained only on the patterns 1 and 4, shown in blue and green (both solid), respectively, the pattern 2 and 3 (dashed) are absent during training. b) RNN is initialized with sine waves yielding to reverberation 2. Upon transitioning into autoregressive mode, the RNN exhibits side dynamics that ultimately guide the neural dynamics towards the trained pattern 1 (blue in a)). c) The trained RNN, initialized with pattern 3, demonstrates side dynamics and a transition towards states aligning with the trained input pattern 4 (green in a)).

## 3 CONCEPTOR-AIDED RECURRENT AUTOENCODER

This section reviews the basics of the conceptor theory and its application to controlling RNNs in the field of reservoir computing. A comprehensive review can be found in Jaeger (2014). We then explain how the framework of conceptors can be combined with backpropagation to guide representation learning in RNNs. Thereby, we are using conceptors during training to constrain the variability of neural dynamics' into a low dimensional geometry.

For the rest of the paper, we will focus on the following simple discrete-time RNN:

$$x(n + 1) = (1 - \alpha)x(n) + \alpha \tanh(W x(n)) + W_{in} u_i(n) + b \tag{3}$$

---

[1]Here, autonomous mode refers to a dynamical system running in the autoregressive mode.

where $W$ is the $N \times N$ internal connectivity matrix, $W_{out}$ is the $M \times N$ output matrix, $W_{in}$ is the $N \times M$ input matrix, $\alpha$ is an $N \times 1$ dimensional leakage vector that governs the update speed of the state and $b$ is a $N \times 1$ bias vector. The RNN is fed with a discrete-time signal $p(n)$, where $p(n) \in \mathbb{R}^M$ and $n > 0$. We will focus on data-sets of two discrete-time patterns $u_1(n)$ and $u_2(n)$ of dimension $M$. These RNNs are known as Leaky Echo-State Networks in the reservoir computing literature (Lukoševičius et al., 2012). Their corresponding autoregressive mode is defined by:

$$x(n + 1) = (1 - \alpha)x(n) + \alpha \tanh(W^* x(n)) + b) \tag{4}$$

where $W^* = W + W_{out}W_{in}$ to regenerate the input signal in the autoregressive mode. Here, the autoregressive mode is created by adding a low-rank component $dim \leq M$ to the internal connectivity matrix used in the input-driven mode. Besides the here used RNN, in the appendix C, we showcase the transfer of our method to the LSTM framework.

### 3.1 CONCEPTORS: CORE THEORY

Conceptors were initially designed to control RNNs with multiple stable and switchable dynamical behaviors. The main challenge of this task is that, while moving the system from the input-driven training in Eq. 1 to the autoregressive mode in Eq. 2, the system is prone to interferences between the trained trajectories. To deal with top-down switchability of behaviors and interferences it was proposed to insert a *matrix conceptor C* in the update equation to project and shield the dynamic:

$$x(n + 1) = C[(1 - \alpha)x(n) + \alpha \tanh(Wx(n) + b)], \tag{5}$$

where $C$ is a soft-projection matrix that preserves some dynamical features of $x$ and discards others. $C$ can be defined by an objective function when considering $x$ as a random variable:

$$\mathbb{E}_x \|Cx - x\| + \alpha^{-2}\|C\|_{\text{fro}}^2 \tag{6}$$

where $\alpha$ is a control parameter called *aperture*. This objective function has a unique analytical solution (Jaeger, 2014) given by:

$$C = R(R + \alpha^{-2}I)^{-}1 \tag{7}$$

where $R = \mathbb{E}_x[x^T x]$ is the $N \times N$ correlation matrix of $x$ and $I$ is the $N \times N$ identity matrix.

The findings presented by Jaeger (2014) can be better understood by examining the singular value decomposition (SVD) of $R$. If we denote the SVD of $R$ as $R = U\Sigma U^T$, then the SVD of $C$ can be represented as $USU^T$, where the matrix's singular values $s_k$ can be expressed as a function of the singular values $\sigma_k$ of $R$: $s_k = \frac{\sigma_k}{(\sigma_k + \alpha^{-2})} \in 0, 1)$. Conceptors can be visualized as ellipsoids within the state space as illustrated in Fig. 2, where the directions and lengths of their axes are determined by the singular vectors and values. To put it simply, $C$ is a soft projection matrix that operates on the linear subspace containing the samples of $x$. For a vector $\tilde{x}$ in this subspace, $C$ behaves like the identity operator: $C\tilde{x} \approx \tilde{x}$. When additional noise or interference $\varepsilon$ orthogonal to the subspace is introduced to $\tilde{x}$, $C$ removes the noise: $C(\tilde{x} + \varepsilon) \approx \tilde{x}$.

### 3.2 CONCEPTORS FOR AUTOENCODING: CONSTRAINING THE VARIABILITY OF NEURAL DYNAMICS GEOMETRY

Conceptors were initially developed for accessing a collection of discrete memories, employing one conceptor per memory to exert top-down control over the dynamics (see Equation 5). We expand their application to encompass a scenario involving a manifold of conceptors, with the discovery of this manifold occurring during the training process.

We will first explain the idea behind the two-shot learning algorithm in plain words before we present the equation of the conceptor regularizer. Driving the RNN with the two data sets respectively generates two distinct conceptors, defining the low dimensional subspace that is characteristic for each input pattern. During the training, we will measure the distance of the conceptors to each other and using a regularization loss enforce them to become closer. Thereby, the network learns to connect the input patterns within network space by generating a continuous manifold as later shown in Fig. 5. After having reached a certain degree of closeness, during inference the network allows to continuously interpolate along conceptor manifold that can be accessed using the continuously interpolatable conceptor $C(\lambda)$. This interpolation scheme is illustrated in Fig. 2.

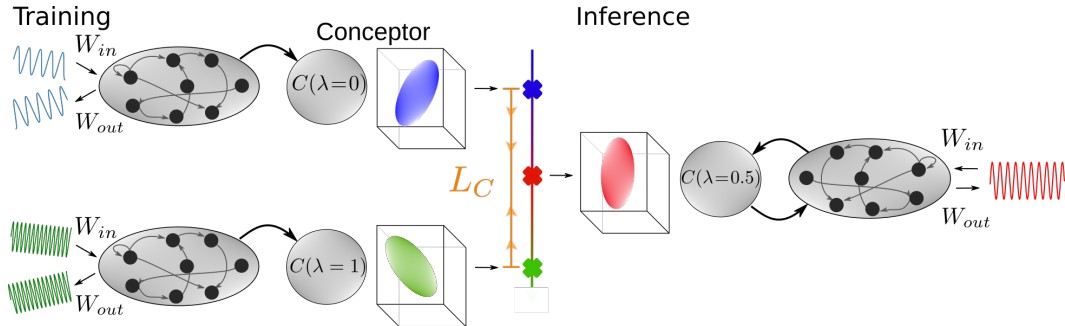

Figure 2: Scheme of the conceptor-aided-recurrent-autoencoder. On the left side the network is trained in the *input driven mode* with distinct temporal patterns blue and green and the conceptor (visualized as an ellipsoid in the state space) is only used for computing the loss $L_C$. After training, the conceptor manifold encodes the two training patterns close to each other. In the inference (right side), a linearly interpolated conceptor is plugged in, and the network generates in the *autoregressive mode* novel intermediary patterns (red) that were not part of the training data.

During training the RNN with BPTT, we employ the conceptor-based constraint using the following loss:

$$L_C = \beta_1 \|C_1 - C_2\|_{fro}^2 + \beta_2 (m_1 - m_2)^2 \tag{8}$$

$$L = MSE(\hat{y}, y) + L_C \tag{9}$$

where $C_i$ is the conceptor computed for pattern $u_i(n)$ and $m_i$ is the network's mean activation vector $m_i = \frac{1}{T} \sum_t x(n)$ when the RNN is driven by pattern $u_i(n)$. The hyperparameters $\beta_1$ and $\beta_2$ balance the cost of reconstruction and the costs associated with the bottleneck. For both experiments presented in the next section, we attached a ablation study of the $(\beta_1, \beta_2)$ in the appendix B.

For clarity, we present the pseudocode of our algorithm in Alg. 1. More details on the implementation of the forward and backward pass, including a derivation of the gradients for all components of the proposed loss function, can be found in Appendix G.

---

**Algorithm 1** Backpropagation through time (BPTT) with conceptor-based regularisation

---

initialize RNN($W, W_{in}, W_{out}, \alpha, b$)
**for** $epoch = 1$ to $n_{epoch}$ **do**
    **for** $u_i$ in $U = [u_1, u_2]$ **do**
        create input and target $u_i, (y_i = u_i(n+1:))$
        **for** $n = 0$ to $N$ **do**
            $x_i(n+1) = (1-\alpha)x_i(n) + \alpha \tanh(W x_i(n) + W_{in} u_i(n+1) + b)$
            $\hat{y}_i(n+1) = W_{out} x_i(n+1)$
        **end for**
        $R = x_i^T x_i$ # compute correlation matrix
        $C_i = R(R + \alpha^{-2}I)^{-1}$ # compute conceptor
        $m_i = mean(x_i)$
        reconstruction loss $L_{rec,i} = \text{MSE}(\hat{y}_i, y_i)$
    **end for**
    conceptor loss $L_C = \beta_1 \|C_1 - C_2\|_{fro}^2 + \beta_2 (m_1 - m_2)^2$
    Loss $L = \sum_{i=0}^{2} L_{rec,i} + L_C$
    compute gradient of $L$ w.r.t. $W, W_{in}, W_{out,\alpha,b}$
    update $W, W_{in}, W_{out}, \alpha, b$
**end for**

---

By employing the conceptor loss $L_C$, we enforce the RNN to encode the two input patterns within a close linear subspace of the network state space $x(k)$. This additional constraint on the available space in the RNN mirrors the concept of a latent space in autoencoder architectures. In situations involving more than two samples, our architecture transforms a nonlinear manifold within the conceptor space into a linear one, effectively introducing a constraint on the variability of the ellipsoids.

It's worth noting that our architecture relies on two types of "compressions". The first type implicitly assumes that each dynamic is confined to a low-dimensional subspace (within an ellipsoid smaller than the entire subspace). The second type constrains the variation of these ellipsoids to a line within the matrix space. This constraint is noteworthy as symmetric positive definite (SPD) matrices are closed under typical matrix interpolation techniques (Feragen & Fuster, 2017). In the appendix F, we give a short discussion of the usage of these metric within the CARAE framework.[2]

Finally, the system can be used in a generative mode by interpolating in conceptor space to control the autoregressive dynamics in Eq. 5 via:

$$C_{\text{interp}}(\lambda) = (1 - \lambda)C_1 + \lambda C_2 \tag{10}$$

## 4 EXPERIMENTS

### 4.1 PATTERN GENERATION

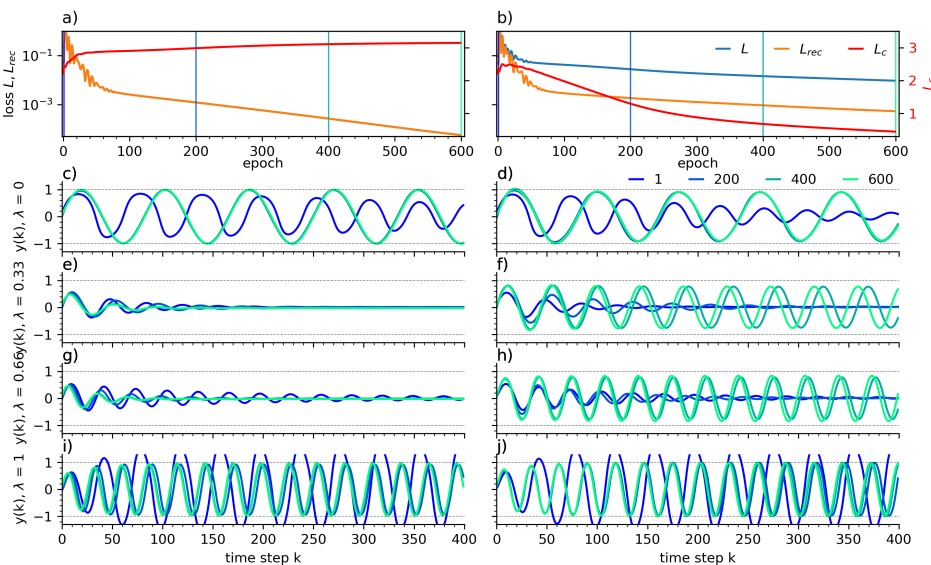

Figure 3: Comparing the interpolation capabilities of an RNN trained on two sine waves with distinct frequency using only the reconstruction loss (left column) and both the reconstruction + conceptor loss (right column). a) Losses over training epochs using only the reconstruction loss $L_{rec}$. b) Losses over epochs using the reconstruction loss (orange) and the conceptor loss (red). Sine wave generation of the RNN in the autoregressive mode after different training epochs ([1,200,400,600], color coded) for different interpolation parameter: c) and d) $\lambda = 0.0$, e) and f) $\lambda = 0.33$, g) and h) $\lambda = 0.66$, i) and j) $\lambda = 1.0$

In the following, we demonstrate the mechanism of the introduced regularisation term on a pattern generation task. Therefore, we have two distinct time series data sets $u_1(n),u_2(n)$ which represent sine waves with different periods $T_1 = 83.8$, $T_2 = 27.9$.

We first train a RNN on a one step ahead prediction only using the reconstruction loss given by the MSE($y,\hat{y}$) and BPTT. As shown in Fig. 3 a), as training progresses, the RNN reduces the reconstruction loss $L_{rec}$ on the two data sets. At several epochs we evaluate the autoregressive retrieval abilities of the RNN by plugging in the conceptors $C(\lambda = 0)$ and $C(\lambda = 1)$. The RNN retrieves the learned sine waves independently as shown in Fig. 3 c) and i). However, as we show in Fig. 3 panels e) and g), using the intermediate conceptors $C(\lambda = 0.33)$ and $C(\lambda = 0.66)$ to enforce intermediate subspaces on the RNN its autoregressive prediction yields unwanted fixed point dynamics

---

[2]We also started to explore different geodesics associated with other metrics on the space of SPD matrices (Feragen & Fuster, 2017). Whereas they could not fix the catastrophic fixed point dynamics on the vanilla RNN shown in the left column of Fig. 3, in combination with the conceptor-based regularisation they can lead to different interpolation paths. However, this goes beyond the aim of this work.

that converge rapidly. Additionally, we monitor in Fig. 3 a) the conceptor loss $L_{rec}$ as defined 8 that estimates the distance between the linear subspace of pattern 1 and 2 in the RNN. As we do not use the conceptor loss in the training, the loss increases and saturates at a higher level. Accordingly, the network encodes the two pattern in linear subspaces that are far from each other. During the interpolation the RNN does not yield stable neural dynamics and hence the output of the network converges to a fixed point.

In contrast, by applying the conceptor regularisation term during training with BPTT we enforce closeness along the conceptor manifold. As shown in Fig. 3 b), we obtain that while reducing the reconstruction loss L, the network is able to reduce the conceptor loss. While reducing the conceptor loss, the RNN starts to generate better interpolations that exhibit intermediary frequencies that lie in between the trained ones. After epoch 600, the network has learned to generalize towards sine waves with untrained frequency. Guided by the plugged in interpolated conceptors, the network can autonomously generate a continuous spectrum of frequencies while being trained on only two different examples.

## 4.2 MOCAP MOTION MODELLING

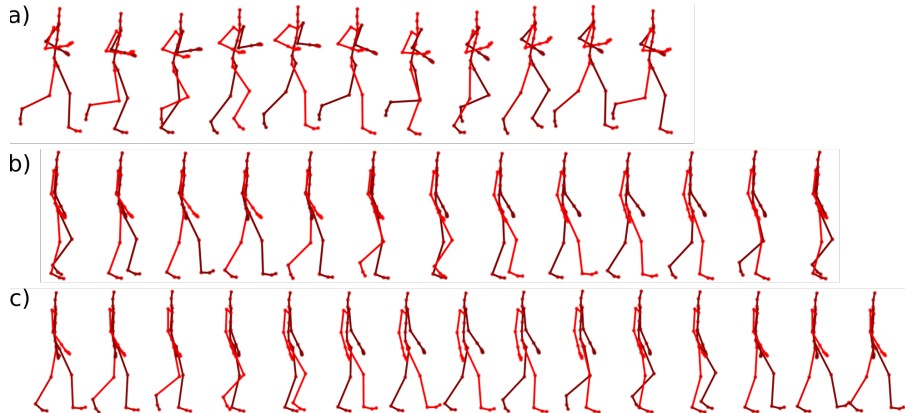

Figure 4: Motion modelling based on the MoCap dataset. The RNN was trained on the temporal patterns of running and walking. In the autoregressive mode it can retrieve both motions as shown in a) and c). Due to enforcing the conceptor based bottleneck during training the RNN and by plugging in a linerly interpolated conceptor it is able to generate intermediate pattern such as shown in b). Each row depicts a single period of each motion pattern at the same sampling rate and with equal horizontal spacing.

In the previous section 4.1, we have shown that based on the conceptor regularisation term a RNN can learn continuous pattern generation while being only trained on two distinct examples. In the following, we use our regularisation to learn more complex patterns based on the MoCap motion capture data set. In particular, we select one example time series for walking (CMU_016_15) and one for running (CMU_016_55) and preprocessed into normalized relative angles following the procedure mentioned in (Jaeger, 2017). The input time series thereby contain 94 dimensions representing the position and angle of the joints of the stick man. As in the previous example, we train our network in the one step ahead prediction. By applying the conceptor-based regularisation as described in 8 the network does not only learn to model the running and walking pattern in the autoregressive mode as shown in Fig. a) and c). Furthermore, in the autoregressive mode we linearly interpolate the plugged in conceptor to guide the RNN towards intermediate subspaces. By doing this, we can generate a continuous spectrum of stable novel motion pattern that interpolate the features such as frequency and posture between walking and running as shown in Fig. 4 panels b).

Thereby, as we show in Fig. 5, the loss of the reconstruction as well as the conceptor loss decrease during training with the two motion patterns. By visualizing the first three principal components of the networks state space in Fig. 5 b), we observe that with increasing training duration within the RNN state space a continuous manifold emerge that connects the linear subspace of the training examples.

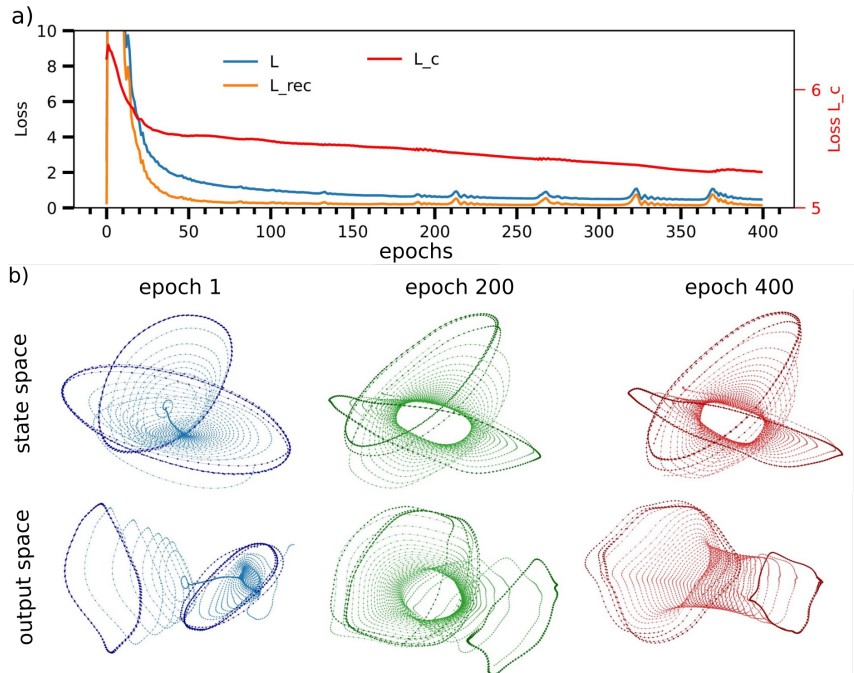

Figure 5: Training a RNN with conceptor-based regularisation on the MoCap datasets for walking (CMU_016_15) and running (CMU_016_55). a) The overall loss (blue 1 ), reconstruction loss (orange) and the conceptor loss (red) over training epochs. b) Three first principal components of the state space $x$ and the output space $y$ after different training epochs evaluated in the autoregressive mode and plugging in linearly interpolated conceptors $C_{\text{interp}}(\lambda), \lambda \in [0, 1]$. The walking and running patterns are highlighted in darker colors. For each epoch of the first row, the walking trajectory corresponds to the horizontal trajectory, while in the second row, it corresponds to the trajectory on the left side.

### 4.3 MoCap motion modeling control

We've seen that CARAE allows us to enforce intermediary dynamical behavior, and now we are considering how to control this intermediary behavior efficiently. In control theory, the aim is to reach a target quickly, stably, and without bias. It's usually very difficult to ensure this for high-dimensional nonlinear systems because arbitrary things can happen when an input is injected. Fortunately, by enforcing a line in control space, we designed a space where control could be ideally done. With a simple gain control, we can check the suitability of the intermediary representation for control in achieving a "speed" control of the transition from walking to running behavior of the agent:

$$C_{\text{interp}}(\lambda_{\text{ctrl}}(n)) = (1 - \lambda_{\text{ctrl}}(n))C_{walk} + \lambda_{\text{ctrl}}(n)C_{run} \tag{11}$$

$$\lambda_{\text{ctrl(n)}} \mathrel{+}= g \cdot \frac{T_{\text{target}} - T_{\text{output}}}{T_{\text{walk}} - T_{\text{run}}} \tag{12}$$

where $T_{target}$ is the target period we want to control and $T_{output}$ is the one currently outputted by the RNN and heuristically estimated.

As we show in Fig. 6, a simple linear gain control on the conceptor line can reliably reach the intermediary modes of behavior without bias and stabilize them. However, a simple linear interpolation creates a bias and can lead to instabilities when asked to extrapolate beyond the observed periods Fig. 6.b).

Notably, our investigation reveals a characteristic within the continuum of trajectories displayed in Fig. 6: the preservation of phase. Here, even during rapid adjustments in speed, the agent maintains phase continuity in its motion. We invite interested readers to explore the supplementary materials, where we present a accompanying video on speed control, showcasing a clear visual presentation of our findings.

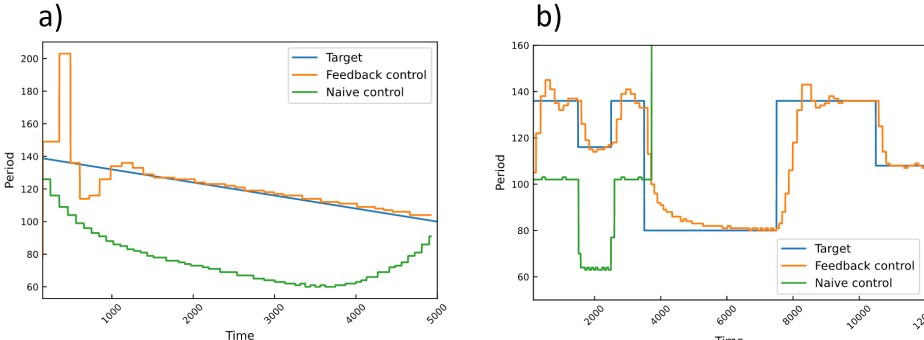

Figure 6: Speed control of the RNN of the locomotion movement by acting on the conceptor line abstracted during training for: a) a ramp target; b) a staircase target (blue) (b). The control is performed by: (green) using linear interpolation $\lambda \in [0, 1]$ of the two learned conceptors (walking $C_1$, running $C_2$); (orange) applying a feedback control on the interpolation parameter $\lambda$.

## 5 CONCLUSION

In this paper, we discuss and show several reasons why dynamics of trained RNN tend to fail in continuous pattern generation when trained in the few-shot and even more dramatic in the two-shot regime. We then introduce a novel conceptor based-regularisation that generates a geometrical interpretable low-dimensional bottleneck within the neural dynamics. In doing so, the regularisation enforces closeness of the linear subspace of the training data. While increasing closeness during training, we observe that within the state space of the network dynamics a continuous manifold emerges. In the inference, running in the autoregressive mode and controlling the RNN along this manifold allows for continuous and phase aligned interpolation. Furthermore, we investigate the efficiency of the learned representations by not only looking at the intermediary representation themselves but studying how controllable they are. Accordingly, we showcase motion speed control via a conceptor during inference using a control loop to steer towards the learned representations. Thus, the conceptor aided recurrent autoencoder becomes a continuously tunable pattern generator while only two data sets containing two distinct temporal patterns are accessible during training. Furthermore, the CARAE framework can be extended to few-shot learning scenarios where more distinct training data set are available, the loss function can be reformulated to include multiple conceptors, each representing a distinct learned temporal pattern as described in the appendix E. This adaptation enables CARAE to handle more complex interpolation scenarios, such as all-to-all interpolation or alignment along a single dimension, or any $k$-dimensional space.

Our results highlight a novel direction in representational learning for time series pattern generation using autoregressive RNNs. By employing the conceptor framework, we enforce geometrical priors that enforce a bottleneck on the neural dynamics during learning. We note that conceptors allow to enforce a bottleneck on the space of trajectories directly. For a periodic signal, this allows the bottleneck to be independent of the initial phase of the signal and the number of periods. This criterion might be seen as analog of "disentangled representation", a classical case study in static autoencoder research. Additionally, we could implement the CARAE framework with various architecture as presented in the appendix C showcasing its general notion. Ongoing and future research is devoted to pushing CARAE to the domain of reinforcement learning and control where agents are embedded in a physical environment (Wagener et al., 2022).

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

## A  HYPERPARAMETERS OF EXPERIMENTS

Here, we present a table with the hyperparameters used for the two experiments 4.1 and 4.2. For the experiments of speed control 4.3 we re-used the Mocap RNN described in the table and used a gain $g = 0.001$.

| Experiments | Sine waves | Mocap |
|---|---|---|
| size RNN ($N$) | 512 | 500 |
| input size ($M$) | 1 | 94 |
| output size ($M$) | 1 | 94 |
| Optimizer | ADAM (default parameters) | ADAM (default parameters) |
| Learning rate $\eta$ | 1e-3 | 0.01 |
| $\beta_1$ | 0.02 | 0.02 |
| $\beta_2$ | 0.002 | 0.1 |
| aperture $\gamma$ | 10 | 10 |

Table 1: Parameters used in the experiments of sine wave interpolation and mocap motion modelling.

## B  QUANTITATIVE MEASURE OF INTERPOLATED TIME SERIES

Our primary objective is to develop an autoregressive recurrent neural network capable of learning a continuum of temporal patterns from just two distinct data sets, such as walking and running, and autonomously generating a range of intermediary patterns. Our approach aims to address the challenge of two-shot learning in dynamical systems, particularly focusing on the generation of intermediate temporal patterns that are not explicitly present in the training data. To quantitatively evaluate the efficacy of our method in inferring intermediary patterns $P$ from two training data sets $A$ and $B$, we propose a similarity metric comprising two independent components. The first component, the Jensen-Shannon divergence $D_{JS}$, a symmetric index of dissimilarity between probability distributions, is calculated as:

$$D_{JS}(A, P) = 1/M \sum_{i=1}^{M} \frac{1}{2}[KL(A_i\|M_i) + KL(P_i\|M_i)] \text{ with } M_i = \frac{1}{2}(A_i + P_i)$$

It is defined as the average of the Kullback-Leibler (KL) divergences of each distribution against their average, ensuring a bounded output between 0 (identical distributions) and 1 (maximally different distributions). This component quantifies the divergence in output state distributions between the intermediary P ($\lambda = 0.5$) and the training sets A and B. Thereby it is calculated per state and averaged across all $M$ output nodes of the recurrent network and the target states. The second component addresses temporal correlations, specifically the autocorrelation, which measures the correlation of a signal with a delayed version of itself as a function of delay. This is essential to capture temporal dependencies in the output states, complementing the state distribution analysis. The autocorrelation difference $D_{acf}$ thereby is computed via:

$$acf(A, \tau) = \left| \frac{1}{K} \sum_{n=1}^{K} \tilde{A}_k \tilde{A}_{k-\tau} \right| \text{ with } \tilde{A} = (A - \mu_A)/\sigma_A$$

$$D_{acf}(A, P) = \frac{1}{T * M} \sum_{i=1}^{M} \sum_{\tau=1}^{T} |acf(A_i, \tau) - acf(P_i, \tau)|$$

Both components, $D_{acf}$ and $D_{JS}$ are normalized and $\in [0, 1]$. The sum of both components related to both training sets and the inferred intermediate pattern constituting the final similarity metric which we refer to as Jensen Shannon and Temporal Correlation difference Composite (JSTCC):

$$JSTCC = \frac{1}{4}(D_{JS}(A, P) + D_{JS}(B, P) + D_{acf}(A, P) + D_{acf}(B, P))$$

Thereby, a lower JSTCC value indicates a more effective combination of information from both training sets, fulfilling our objective of achieving meaningful interpolation between the two training

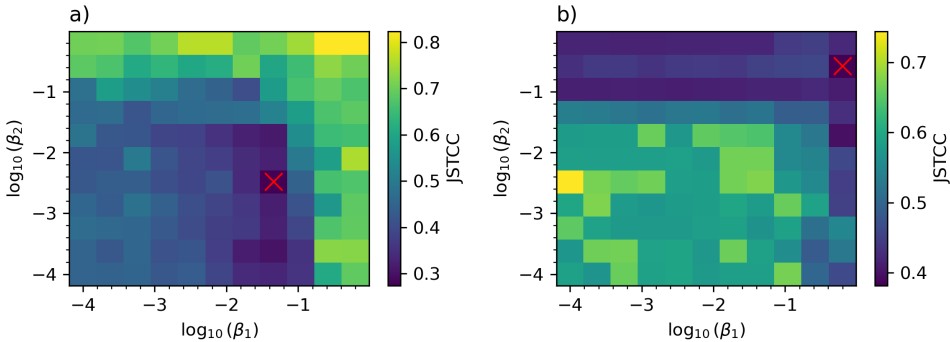

Figure 7: a) Ablation study of the loss function hyperparameters $\beta_1$ and $\beta_2$ of the CARAE framework evaluated using the Jensen-Shannon and temporal correlation difference composite (JSTCC) for generating intermediary sine wave patterns. b) Ablation study of the loss function's hyperparameter for interpolation of the MoCap data set. In both the red cross marks the lowest JSTCC found in the parameter range, respectively. The JSTCC is averaged over 5 random initial conditions of the RNN

data sets. The minimum reachable value of JSTCC is thereby further depending on the similarity of the two training data sets.

In the following, we study the ablation of the regularization parameters $\beta_1$ and $\beta_2$ for the sine wave and the mocap interpolation tasks. We find clear dependence of the JSTCC metric regarding the variation of the parameters that further differs between the two tasks. For the shown tasks, both of the parameters $\beta_1$ and $\beta_2$ are important for yielding sufficiently interpolated solutions. Thereby, we find that the loss term related to the mean of the activation (weighted by $\beta_2$) in Eq. 21 in both task can be chosen smaller than the loss related to the distance of the conceptors (weighted by $\beta_1$). By comparing the tasks, we observe that the sine wave interpolation requires less strong regularization compared to the mocap data interpolation. While this could be related to the much more complex temporal profile of the mocap data set time series, more research is needed to fully reveal the relationship of the regularization parameters and the generated interpolation.

## C    EXTENDING CARAE TO OTHER ARCHITECTURES

In the following, we extend the CARAE framework towards the usage combined with a long-short term memory (LSTM) cell. Thereby, the LSTM replaces the recurrent neural network as given in Eq. 3. We implement LSTM cell mathematically described by the following set of equations:

$$i = \sigma(W_{ii}x + W_{hi}h + b_{hi}) \qquad\qquad f = \sigma(W_{if}x + W_{hf}h + b_{hf})$$
$$g = \tanh(W_{ig}x + W_{hg}h + b_{hg}) \qquad o = \sigma(W_{io}x + W_{ho}h + b_{ho})$$
$$c' = f * c + i * g \qquad\qquad\qquad h' = C(\lambda)(o * \tanh(c'))$$
$$y = W_{out}h'$$

where $x$ is the input, $h$ is the output of the previous time step, and $c$ is the memory. The conceptor $C(\lambda)$ is applied during the generation of the output state $h$ of the LSTM cell. Accordingly, only minimal changes are needed to adapt our framework towards the LSTM. We test the performance of the LSTM on the intermediary pattern generation while relying on the proposed loss function Eq. 21 during training. In the inference, as shown in Fig. 8 c), we find that LSTM learns to generate to interpolate between the two patterns given during training and reveals sine wave pattern at an intermediary frequency. In Fig. 8 d), we further observe the learning as the loss function defined in Eq. 21 reduces along the epoch while similar the JSTCC reduces at the same time.

Furthermore, we extend the CARAE framework towards a static feed-forward network. Therefore, we employ a vanilla two-layer deep $\tanh$ network that is feed by a moving window with a context

length K along the time series. The equations of this model are given as follows:

$$x_1 = C \tanh(W_{in}u + b), \qquad x_2 = \tanh(W_{12}x_1 + b), \qquad y = W_{out}x_2,$$

$$C = \begin{cases} \mathbb{1} \text{ during training} \\ C(\lambda) \text{ during inference} \end{cases}$$

Thereby, the conceptor is computed based on the activation of the neurons in the first layer along the input sequence $u(k)$. During inference, the conceptor acts as a linear projection on the first layer activation onto a low-dimensional space before the states flow trough the second layer. We find that the feed forward network with a context length of $K = 20$ inputs and running in an autoregressive mode is able to interpolate the sine wave patterns based only on the two training as shown in Fig. 8 a). Similar to the LSTM-based CARAE discussed above, we find that loss and JSTCC reduce at the same time along the training epoch.

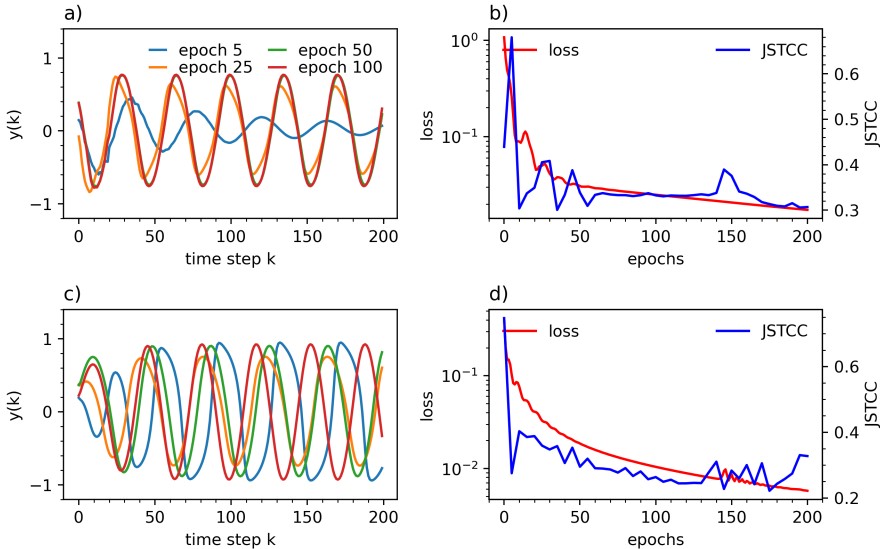

Figure 8: a) Inferred time series at $\lambda = 0.5$ from a feed-forward neural network trained on two distinct sine waves at different epochs (color-coded) using an adapted version of the CARAE framework for non-recurrent networks. b) Loss over epochs and JSTCC measure applied on the inference the feed-forward neural network. c) Inferred time series at $\lambda = 0.5$ from a LSTM network trained on two distinct sine waves using the CARAE framework at different epochs (color-coded). d) Loss over epochs and JSTCC measure applied on the inference the LSTM-based CARAE.

To summarize, in this section we transferred the ideas of CARAE framework towards two other architecture, namely a LSTM-based CARAE and a static feed-forward network that relies on the CARAE loss. Both architecture introduce the ability of generating intermediary pattern in a two-shot learning regime. Thereby, we showcase the generality of the CARAE ideas beyond leaky-integrator networks used in the main text and towards various deep learning frameworks that run in an autoregressive mode.

## D COMPARING CARAE TO CLASSICAL RECURRENT AUTO-ENCODER (RAE)

In this section, we contrast our CARAE approach with traditional Recurrent Auto-Encoder (RAE). In these architectures, an encoder RNN encodes the input time series $u$ into a vector $z(n_{encoder})$:

$$x(n + 1) = RNN_{encoder}(x(n), u(n))$$

$$z(n_{encoder}) = W^{encode}x(n_{encoder})$$

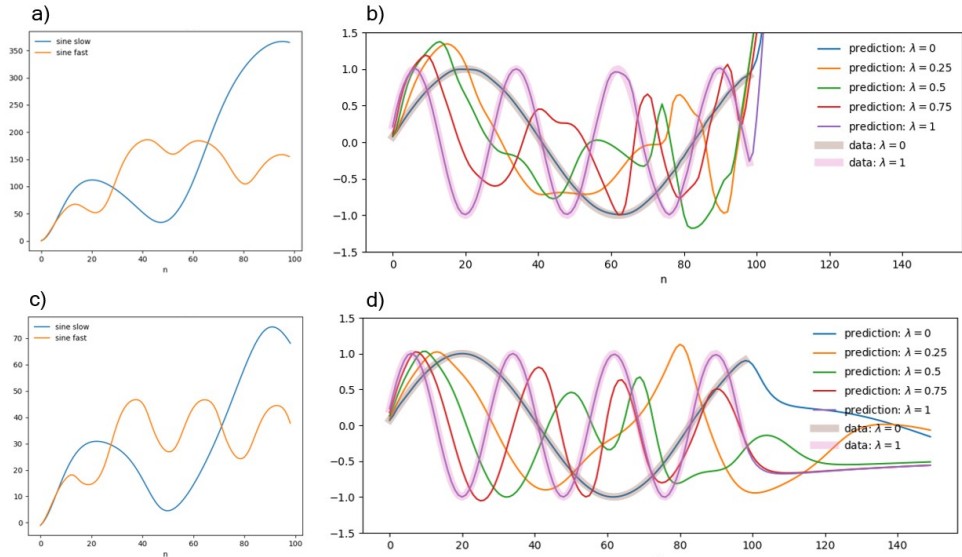

Figure 9: Sine waves auto-encoding with an RAE. a) Output of the encoder RNN ($z_((n)$) for two input sine waves with $\beta = 0$. b) Output of the decoder RNN for $\beta = 0$ and various values of $\lambda$ (color-coded). c) Output of the encoder RNN ($z_((n)$) for two input sine waves with $\beta = 1$.. d) Output of the decoder RNN for $\beta = 1$. and various values of $\lambda$ (color-coded). For the decoding figures b) and c), the two input sine waves used during training are displayed in transparent color. Also, the system is only trained to decode for 100 steps; any steps after are out-of-distributions.

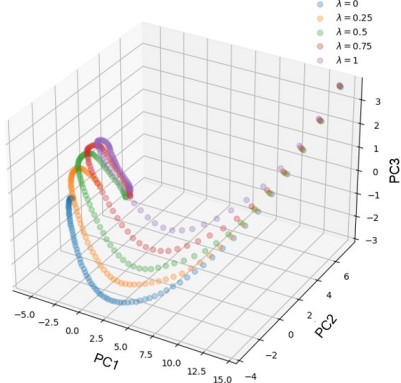

Figure 10: Principal components of the state of the RAE decoder when regenerating sine waves for different values of $\lambda$. The vertical axis correspond to the different values of $\lambda$.

where $n_{encoder}$ is time steps indicating the end of the input, and $RNN_{encoder}$ is based on Eq. 4. Then, a decoder RNN decodes (also based on Eq. 4) the vector to reconstruct the input.

$$x(n+1) = RNN_{decoder}(x(n), z(n_{encoder}))$$
$$y(n) = W^{decoder} x(n)$$

where the latent code $z$ is sent as a constant input to bias the decoding to regenerate the input through $y$.

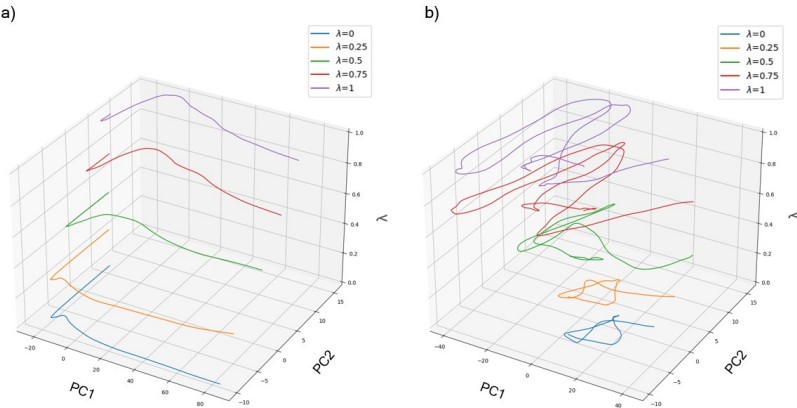

Figure 11: Principal components of the state a) and output b) of the RAE decoder when regenerating walking and running from Mocap data and its interpolation for different values of $\lambda$. $\beta = 10$.

We can compare this classical approach to the CARAE mechanism presented in the paper by adapting the loss function to:

$$L_C = \beta \|z_1 - z_2\|_{fro}^2$$
$$L = MSE(\hat{y}, y) + L_C$$

where $z_1$ and $z_2$ are the codes of the two examples in the two-shot learning task. At test time, we interpolate the latent code by using:

$$z_{interp}(\lambda) = (1 - \lambda)z_1 + \lambda z_2$$

In an experiment analog to section 4.1 with sine waves, the RAE effectively encoded and decoded inputs through a one-dimensional bottleneck (Fig. 9). Without further compression in the latent space ($\beta = 0$), the system overfits the training data without learning the parametric family. Conversely, introducing compression reveals the reconstruction of the parametric family for about 50 initial steps. However, a notable limitation emerges as the interpolation quality deteriorates towards the end of the decoding process, evident even before reaching the out-of-distribution segment. This degradation suggests that the RAE architecture lacks the necessary inductive bias for learning periodic patterns, as observed in the states of the decoder RNN (Fig. 10) which fail to exhibit periodic cycle attractors. This limitation is further highlighted by the RAE's inability to learn the periodic aspect of a fast sine wave despite training on multiple successive periods.

The same qualitative effects of RAE are observed when applied to the Mocap two-shot learning task. The RAE manages to regenerate approximately one period of walking or running behavior (Fig. 11 b)). The system's failure to learn periodicity is again evident from the trajectory shapes in the state of the decoder RNN (Fig. 11 a)).

In summary, while the RAE is capable of basic encoding and decoding functions, its architecture shows significant limitations in learning and reproducing periodic patterns, as demonstrated in both sine wave and Mocap data experiments. Thereby, it struggles with overcoming fixed point dynamics as outlined in section 2. In contrast, our CARAE model, by specifying a bottleneck on the geometry of the neural trajectories, is much more adapted to periodic pattern and overcomes the fixed point dynamics at intermediary interpolation, rendering it a strong framework for locomotion data.

# E    EXTENDING CARAE TO MULTIDIMENSIONAL BOTTLENECKS AND MORE THAN TWO TIME-SERIES.

While the focus of our work was on two-shot learning with a one-dimensional bottleneck we present here how to extend CARAE to multidimensional bottlenecks and an arbitrary number of time-series.

1. **Define a low-dimensional subspace:** Select a metric on the space of matrices. Select $2m$ orthogonal conceptor, where $m$ is the desired dimension for the bottleneck.

2. **Augment the loss with a projection cost:** Augment the reconstruction with a loss that penalizes when a conceptor is outside the low-dimensional subspace in conceptor space.

$$L = \sum_{i \in D_{train}} MSE(y_i, \hat{y}_i) + \beta ||C_i - Proj_C(C_i)||$$

where $D_{train}$ is the training set and $y_i$ and $C_i$ are the time series reconstructed and the conceptor associated with the sample $y_i$. $Proj_C()$ is the linear operator that projects a matrix into the $m-$dimensional linear subspace of the conceptor space. An additional cost could also be added to further compress by reducing the variance along the axes of the $m-$dimensional subspace.

3. **Train the system with the generalized cost function:** Then the whole system can be trained similarly to the two-shot setting.

Note that in this example, the latent conceptor space is defined at initialization and is not changing, but we could also easily imagine backpropagating the conceptors that define it during training.

# F  EXPLORING SPD MATRIX INTERPOLATION TECHNIQUES DURING INFERENCE

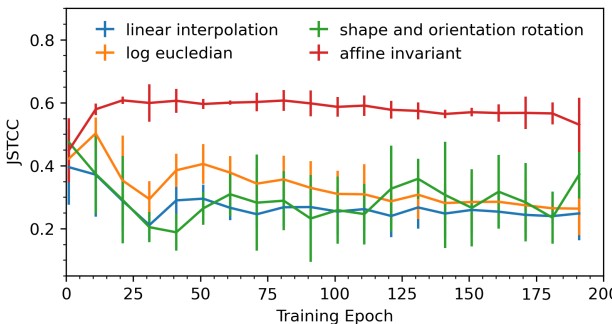

Figure 12: JSTCC metric for the inference of a CARAE trained on sine wave prediction for a intermediary conceptor $C(\lambda = 0.5)$. Four different interpolation techniques for symmetric positive definite matrices are used to generate the intermediary conceptor (color-coded). The JSTCC is averaged over 5 different random seeds.

The CARAE framework relies on the interpolation of two conceptors that are defined as symmetric positive definite (SPD) matrices. For such SPD matrices there exist various forms of interpolation that take two SPD matrices and convert them into another intermediary SPD matrix. In the main text, we employ a linear distance measure of the conceptors for the CARAE loss function given in Eq. 21 and in the inference a linear interpolation of the conceptor is used for the control of the CARAE. The reason for this design decision, is the computational efficiency of both, the linear distance and interpolation. Other metrics and techniques such as log-euclidean, affine invariant and the shape and orientation rotation (SAOR) metric Feragen & Fuster (2017) rely on applying the matrix exponential and matrix logarithm on the SPD matrices. This is not only computationally much more extensive but further the computation of these metric can introduce instabilities that might harm and even break the training process. Whereas, we therefore do not employ them during the training, in this section, we evaluate their application in the inference and interpolation process. Therefore, we train a CARAE model based on a leaky-recurrent neural network on two sine wave pattern and evaluate the performance of the interpolation of the model during various time steps along the training process using these four interpolation techniques. We obtain in Fig. 12, that while optimizing the linear distance between the conceptors in the training, we obtain a decreasing JSTCC during the linear, log-euclidean and SAOR interpolation technique. Whereas the SAOR reaches to lowest JSTCC of all 4 techniques we obtain strong variances of the JSTCC of the SAOR technique

along the training, indicating is instabilities. After 200 epoch the linear interpolation saturates and reaches with a small variance a low JSTCC level. The affine invariant interpolation technique overall does not yield low JSTCC and the performance does not seem to be affected strongly be the training of the CARAE.

In Fig. 13, we present the interpolated sine waves for the 4 different techniques at four different epoch: a) 1 b) 40 c) 120 d) 200. We obtain, that the linear and log-euclidean interpolation techniques improve the generalization of the CARAE with increasing training epoch as suggested by Fig. 12. Interpreting the results of both figures, we suggest that based on the closeness of these techniques the results appear similar whereas the linear interpolation seems to be slightly better which might be due to the reliance on the linear loss function. The SAOR interpolation show promising results already after 40 epoch as shown in Fig. 13 b), however further training does not improve the interpolation using this technique. In contrast, the SOAR results in fixed point dynamics for longer training as shown in Fig. 13 panel c) and d). This might be overcome by using the SOAR distance metric in the training which as mentioned above might introduce mathematical instabilities and might be a subject of further investigation. The affine invariant interpolation technique, as suggested by Fig.12, yields to fixed point dynamics for all training epochs.

To summarize, in this section, we give a first evaluation of different interpolation techniques of SPD matrices that could be used together with the CARAE framework. Here, we restrict the application to the inference part only, while further research might use them in the loss function as well. Our results indicate that the linear and log-euclidean interpolation technique yield quite similar results, whereas the other two, SAOR and affine invariant technique suggest a diverging alignment related to the linear distance based CARAE training.

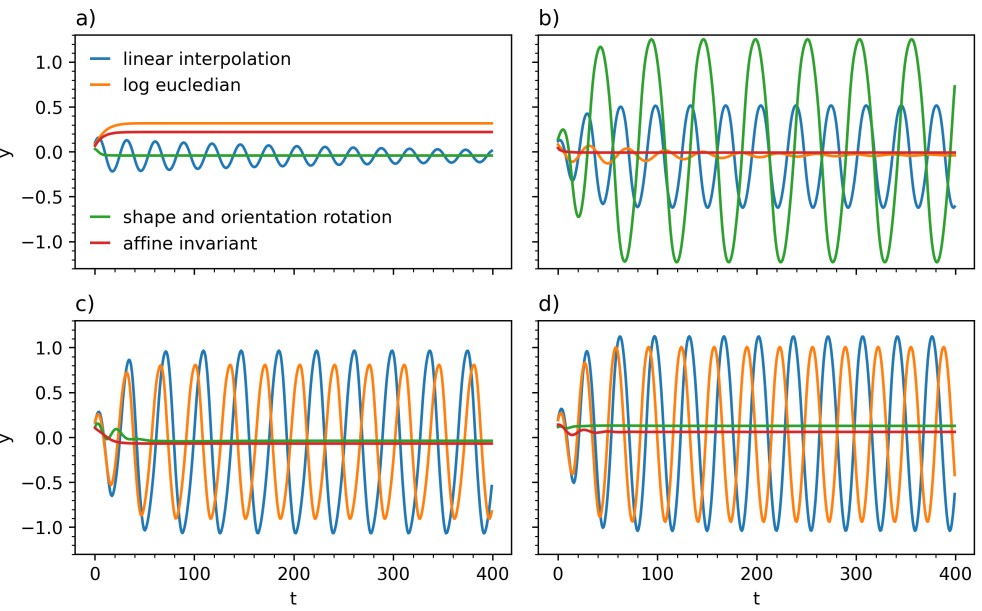

Figure 13: Inference of intermediary patterns ($\lambda = 0.5$) for the sine wave task using different interpolation techniques (color-coded) at 4 different epochs during the training: a) 1 b) 40 c) 120 d) 200)

## G DERIVATION OF BACKPROPAGATION THROUGH TIME THROUGH CONCEPTOR

For backpropagation through time through conceptor, we use a RNN in input-driven mode, as given by Eq. 1 and 4.

### G.1 FORWARD PASS: LOSS COMPUTATION

In our two-shot learning setup, we pass a mini-batch of two $M$-dimensional time series into the network, $\mathbf{u}_n = [u_n^0 \ u_n^1]^T \in \mathbb{R}^{2 \times M}$, such that our forward pass is given by:

$$\mathbf{x}_n = (1 - \alpha)\mathbf{x}_{n-1} + \alpha \tanh\left(W\mathbf{x}_{n-1} + W_{in}\mathbf{u}_n + b\right) \tag{13}$$

$$\mathbf{y}_n = W_{out}\mathbf{x}_n + b_{out} \tag{14}$$

where $\mathbf{x}_n \in \mathbb{R}^{2 \times N}$ is the mini-batch of state vectors, $\mathbf{y}_n \in \mathbb{R}^{2 \times M}$ is the mini-batch of output vectors, and $n \in \{1, \dots, T\}$ indexes the sequence length. Note that weight matrices are batch-multiplied and the bias vectors are broadcasted such that the addition is element-wise across the batch dimension.

After the forward pass, we use the collected state matrix for each of the two input patterns $X_i = [x_1^i \dots x_T^i]^T \in \mathbb{R}^{T \times N}$ where $T$ is the total sequence length of the input signals. We compute the conceptor matrix $C_i$ for each input pattern as:

$$R_i = X_i^T X_i \tag{15}$$

$$C_i = R \left( R + \frac{\mathbb{1}_N}{\gamma^2} \right)^{-1} \tag{16}$$

where $R_i$ is the state correlation matrix, $\mathbb{1}_N$ is the $N$-dimensional identity matrix, $\gamma$ is the aperture of the conceptor, and $^{-1}$ denotes the matrix inverse.

We further compute the RNNs mean activation vectors $m_i$ as:

$$m_i = \frac{1}{T} \sum_{k=1}^{T} x_k^i \tag{17}$$

We then compute the conceptor-based regularization terms for the loss function according to equations 8 and 9:

$$L_C = \beta_1 \|C_1 - C_2\|_{fro}^2 \tag{18}$$

$$L_m = \beta_2 (m_1 - m_2)^2 \tag{19}$$

where $\|.\|_{fro}$ is the Frobenius matrix norm.

The reconstruction loss is computed as the mean squared error between the network output $y_n^i$ and the ground truth $\hat{y}_n^i$, yielding the final loss function:

$$L_y = \frac{1}{2T} \sum_{i=1}^{2} \sum_{k=1}^{T} (y_n^i - \hat{y}_n^i)^2 \tag{20}$$

$$L = L_C + L_m + L_y \tag{21}$$

### G.2 BACKWARD PASS: GRADIENT COMPUTATION

To update the weights and biases of the RNN, we need to compute the gradients with respect to the output weights $W_{out}$, the output bias $b_{out}$, the recurrent weights $W$, the recurrent bias $b$, and the input weights $W_{in}$. We decompose the gradient according to the loss decomposition shown in Eq. 21:

$$\frac{\partial L}{\partial \theta} = \frac{\partial L_C}{\partial \theta} + \frac{\partial L_m}{\partial \theta} + \frac{\partial L_y}{\partial \theta} \tag{22}$$

Here we focus on the computation of the gradients for the novel loss components $L_C$ and $L_m$ without further deriving the gradients for $L_y$.

The gradient of the loss with respect to $W_{out}$ simplifies to the gradient of the mean squared error loss with respect to $W_{out}$ because neither $L_C$ nor $L_m$ depend on $W_{out}$:

$$\frac{\partial L}{\partial W_{out}} = \underbrace{\frac{\partial L_C}{\partial W_{out}}}_{=0} + \underbrace{\frac{\partial L_m}{\partial W_{out}}}_{=0} + \frac{\partial L_y}{\partial W_{out}} \tag{23}$$

$$\frac{\partial L}{\partial W_{out}} = \frac{\partial L_y}{\partial W_{out}} \tag{24}$$

Analogously, the gradient of the loss with respect to $b_{out}$ simplifies to the gradient of the mean squared error loss with respect to $b_{out}$.

Clearly, the gradient of the loss with respect to $W$, $b$, and $W_{in}$ cannot be simplified in the same way. We proceed to derive the gradient of the auxiliary loss term $L_m$ with respect to $W$ and note that the derivation with respect to $b$ and $W_{in}$ is analogous:

$$\frac{\partial L_m}{\partial W} = \sum_{i=1}^{2} \frac{\partial L_m}{\partial m_i} \frac{\partial m_i}{\partial W} \tag{25}$$

$$= \sum_{i=1}^{2} \sum_{k=1}^{T} \frac{\partial L_m}{\partial m_i} \frac{\partial m_i}{\partial x_k^i} \frac{\partial x_k^i}{\partial W} \tag{26}$$

$$= \sum_{i=1}^{2} \sum_{k=1}^{T} \sum_{l=1}^{k} \frac{\partial L_m}{\partial m_i} \frac{\partial m_i}{\partial x_k^i} \frac{\partial x_k^i}{\partial x_l^i} \frac{\partial x_l^i}{\partial W} \tag{27}$$

$$= \sum_{i=1}^{2} \sum_{k=1}^{T} \sum_{l=1}^{k} \frac{\partial L_m}{\partial m_i} \frac{\partial m_i}{\partial x_k^i} \left( \prod_{j=l+1}^{k} \frac{\partial x_j^i}{\partial x_{j-1}^i} \right) \frac{\partial x_l^i}{\partial W} \tag{28}$$

The gradient of the conceptor loss $L_C$ with respect to $W$ can be derived as:

$$\frac{\partial L_C}{\partial W} = \sum_{i=1}^{2} \frac{\partial L_C}{\partial C_i} \frac{\partial C_i}{\partial W} \tag{29}$$

$$= \sum_{i=1}^{2} \sum_{k=1}^{T} \frac{\partial L_C}{\partial C_i} \frac{\partial C_i}{\partial R_i} \frac{\partial R_i}{\partial X_i} \frac{\partial X_i}{\partial x_k^i} \frac{\partial x_k^i}{\partial W} \tag{30}$$

$$= \sum_{i=1}^{2} \sum_{k=1}^{T} \sum_{l=1}^{k} \frac{\partial L_C}{\partial C_i} \frac{\partial C_i}{\partial R_i} \frac{\partial R_i}{\partial X_i} \frac{\partial X_i}{\partial x_k^i} \frac{\partial x_k^i}{\partial x_l^i} \frac{\partial x_l^i}{\partial W} \tag{31}$$

$$= \sum_{i=1}^{2} \sum_{k=1}^{T} \sum_{l=1}^{k} \frac{\partial L_C}{\partial C_i} \frac{\partial C_i}{\partial R_i} \frac{\partial R_i}{\partial X_i} \frac{\partial X_i}{\partial x_k^i} \left( \prod_{j=l+1}^{k} \frac{\partial x_j^i}{\partial x_{j-1}^i} \right) \frac{\partial x_l^i}{\partial W} \tag{32}$$

where the derivation for $W_{in}$ and $b$ is analogous.

Notably, in computing the term $\frac{\partial C_i}{\partial R_i}$, the gradient flows through a matrix inversion:

$$\frac{\partial C_i}{\partial R_i} = \frac{\partial}{\partial R_i} R_i \left( R_i + \frac{\mathbb{1}_N}{\gamma^2} \right)^{-1} \tag{33}$$

As this is a derivative of a conceptor matrix $C_i \in \mathbb{R}^{N \times N}$ with respect to the correlation matrix $R_i \in \mathbb{R}^{N \times N}$, the resulting partial derivative is expressed as a four-dimensional tensor $\frac{\partial C_i}{\partial R_i} \in \mathbb{R}^{N \times N \times N \times N}$. This can be simplified to a matrix by computing the derivative of the scalar conceptor loss directly, $\frac{\partial L_C}{\partial R_i} \in \mathbb{R}^{N \times N}$.

# H LYAPUNOV SPECTRUM OF CARAE VS. VANILLA RNN

To quantify the qualitative change of the dynamics within the trained RNNs either using the CARAE framework or vanilla RNN with BPTT, we compute their Lyapunov spectrum. Conceptually, the Lyapunov spectrum gauges the stability of trajectories along an attractor. The computation involves the following steps: first, the generation of an orbit along the attractor of the RNN, $x(n)$; second, the evaluation of the Jacobian at each point along the orbit, $J(x(n))$; and finally, the evolution of orbits of infinitesimal perturbations, $\mathbf{p}_k(n)$, along the time-varying Jacobian as

$$\dot{p}_k(n) = J(x(n)) p_k(n) \tag{34}$$

Along these orbits, the direction and magnitude of $p_k(n)$ change based on the linearly stable and unstable directions of the Jacobian $J(x(n))$. To capture changes along the orthogonal directions, following each time step of evolution, we organize the perturbation vectors into a matrix

$[p_1(n), p_2(n), \ldots, p_k(n)]$, and perform a orthonormalization based on the QR algorithm. Thus, $\tilde{p}_1(n)$ eventually aligns with the least stable direction, $\tilde{p}_2(n)$ with the second least stable direction, and $\tilde{p}_k(n)$ with the most stable direction. The rate of divergence or convergence along the perturbation directions is given by the so-called Lyapunov exponents (LE).

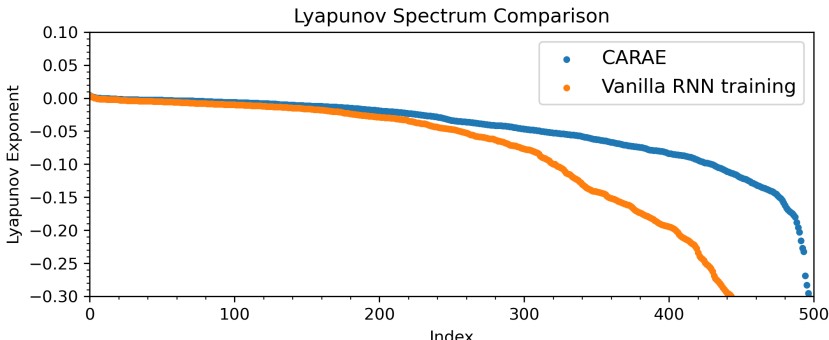

Figure 14: Lyapunov Spectrum of a 500-neuron RNN trained on the MoCap dataset of running and walking using the CARAE framework (blue) and a vanilla RNN (orange).

In Fig. 14, we show the LE spectrum of two differently trained RNNs: one trained in a vanilla BPTT fashion and the other trained using our here introduced CARAE framework. We observe that, the spectrum from the CARAE contains more LE close to zero. A LE close to zero hereby indicates a direction in the state space in which the dynamics neither diverge nor converge. In Smith et al. (2022) such LEs equal zero are connected to learned abstraction in RNNs. Whereas we can identify a qualitative change in the Lyapunov spectra between the two training methods, more work is needed to connect e.g. the direction of the emerging manifold in the CARAE framework shown in Fig. 5 b) with the Lyapunov spectrum of the RNN.

## I   QUANTIFICATION OF INTERPOLATION SUCCESS FOR MOCAP

To quantify the interpolation success, we leverage the ideas of Huguet et al. (2022). After training we query CARAE with an intermediary pattern, we infer the latent variable (for CARAE, a conceptor) and then assess the capacity to continue the pattern.

For MoCaP, we used the pattern (CMU0 16 35), a jogging pattern, roughly an intermediary between walking and running.

**Details of the test:** the pattern (CMU0 16 35) is entirely presented, corresponding roughly to two periods of jogging (161 steps). After computing the conceptor, we continue the dynamic from the last state of the cueing period and plug the conceptor into the RNN to use it generatively for 80 steps (one period of jogging).

The lambda for the CARAE is obtained by projecting the conceptor inferred onto the conceptor line defined by the walking and running conceptor.

To deal with the phase dependence of the MSE, we first phase align the generated pattern (by searching for the MSE best match) to the ground-truth (pattern CMU 16 35). For the linear interpolation baseline, we also search for the best phase alignment between walking and running before interpolating.

The tests are done over 80 time steps (one periods of jogging)

We see in Fig. 15 that the CARAE is better than linear interpolation at matching the jogging pattern for both metric mse and jstcc.

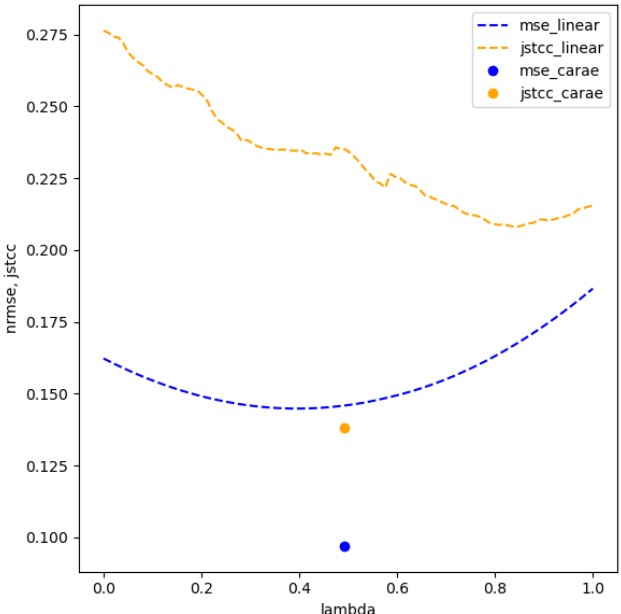

Figure 15: Quantitative measure of interpolation: we compare CARAE and a linear interpolation baseline between walking and running to the intermediary patter of jogging. See text for details.

