# OpenReview forum: "Two-shot learning of continuous interpolation using a conceptor-aided recurrent autoencoder"
_ICLR.cc/2024/Conference — Submitted to ICLR 2024_

### Official Review · Reviewer_PFFz · 2023-10-30

**Soundness:** 2 fair
**Presentation:** 2 fair
**Contribution:** 2 fair
**Rating:** 3
**Confidence:** 4

**Summary:**

This paper deals with analysis of time series. It is proposed to design a generator of such time series from limited observation data.Those data being limited in the sense that only two patterns type are considered as available. The aim of this paper is then to be able to generate 'intermediate' patterns. This is performed through the use of recurrent auto-encoder with a conceptor-based regularization. This conceptor-based regularization is based on two folds: (1) ensuring a close representation of patterns and (2) ensuring close conceptors for each pattern.
From this trained recurrent auto-encoder, a linear interpolation of conceptors is proposed to generate intermediate patterns.

Experimental results are reported on sine-wave patterns and MoCap motion modelling.

**Strengths:**

The proposed method allows to provide a recurrent auto-encoder generator that is able to generate continuous type of patters although being trained on two patterns.

**Weaknesses:**

The proposed controlled recurrent auto-encoder allows to be able to generate continuous patterns evolution between two initial pattern of time series. However this is quite hard to assess the benefits of the proposed approach since no comparison to other technique is proposed nor any quality metrics. Only subjective evaluation is reported.

References to equation and figures needs to be revisited since when cited it is often omitted to mention if it refers to a table, image or other.

As quickly mentioned in footnote of page 6, SPD matrices are to be associated to a specific manifold. So linear interpolation used in equation 10 could be discutable. But also the distance metric used in equation 8. It would be interesting to consider also here some classical metrics and interpolation technique of SPD matrices to see the impact of distance metric and interpolation technique there. See also [1,2] for additional paper on SPD matrices metrics.

No discussion on the impact of $\beta_1, \beta_2$ parameter is provided. What about for exemple using only pattern proximity (e.g. $\beta_1=0$) ?

Details on backward pass in appendix B.2 is quite useless when considering Deep Learning frameworks that automaticaly compute back-propagation path (e.g. TensorFlow, Pytorch).

[1] Huang et al, Log-Euclidean Metric Learning on Symmetric Positive Definite Manifold with Application to Image Set Classification,
Proceedings of the 32 nd International Conference on Machine Learning, Lille, France, 2015. JMLR: W&CP volume 37, 2015

[2] Vemulapalli et al, Riemannian Metric Learning for Symmetric Positive Definite Matrices

**Questions:**

1. Could there be objective metric to evaluate the performance of proposed approach? For example what about having a 3rd pattern to be used for evaluation?
2. What about the impact of $\beta_1, \beta_2$ hyper-parameters? Could there be a specific ablation study?
3. How does the proposed technique performs with respect to other approaches?

---

> ### Author Response · Authors · 2023-11-19
> **Answer to Reviewer**
>
> Thank you for your critical feedback. We have made several amendments to our manuscript to address your concerns, particularly regarding the evaluation methodology.
>
> We agree with  the need for objective evaluation metrics, and we have devised a quantitative metric in the revised manuscript. In this context, we would not like to use a third pattern to evaluate our method as the interpolated temporal patterns are based on probabilistic interpolation related to the training data. A third example of a parametric family could contain features that are absent or only weakly present in the training data and therefore can not be inferred by the CARAE mechanism. Using such third patterns for evaluating our method could lead to false understanding of what our method aims to achieve. To quantitatively measure the performance of the interpolated temporal patterns we have developed a similarity metric specifically tailored to our task. This metric provides a probabilistic measure and a temporal correlation measure of the generated patterns' fidelity to the training data, addressing the subjective nature of our initial evaluations. Further details on this metric, including its formulation and application, are now included in the appendix and referenced in the manuscript. Finally, we would also like to note that auto-encoders are often judged on their output re-generation with subjective judgments [1]. Motion modeling also [2] because one of their main goals is to develop plausible motion for simulated characters.
> To provide further clarity on the role of conceptor loss and its regularization term, we included in the appendix an ablation study of the regularization parameters beta_1 and beta_2. This addition aims to elucidate the identifiable effects of conceptor loss on controlled neural networks, also illustrating the applicability of the newly introduced evaluation metrics. Both regularization parameters show different influences for the two evaluated tasks. Thereby, we find that the usage of both is needed to significantly improve the generalization during the interpolation.
>
> In response to your remarks on SPD matrices, we would like to discuss our choice of the linear interpolation. We also discussed alternative metrics like log-euclidean and interpolation methods for SPD matrices, as mentioned in the referenced literature. However despite its simplicity, the linear interpolation approach preserves the SPD matrix properties [3], ensuring that the interpolated conceptors are still valid soft projection matrices. Furthermore, we decided to focus on linear interpolation due to its computational efficiency. Other metrics include the usage of matrix logarithm and exponential that are computationally expensive for the forward and backward-pass and could potentially introduce instabilities in the training.  Furthermore, these functions are often not implemented in standard deep learning libraries making their usage more complicated. Nevertheless, as it is computationally more feasible, we tested the usage of the log-euclidean and the shape-and-orientation (SAO) metric during the inference while relying on the linear notion for the training. The log-Euclidean and SAO metrics are developed to tackle known challenges in statistical operations involving semi-definite matrices. One such problem is the "flattening effects" [4]. This refers to the deformation and shrinkage of singular values that occur during linear interpolation. However, in our study, we found that these metrics did not significantly impact the inference process. Whereas this might be related to the particular task at hand we agree and mention in the paper that a more rigorous investigation of the influence of the metric might further improve our results. Due to the limited time, we started this investigation of the metrics for inference only and attached an initial discussion to the appendix as section F.
>
> Minor remarks:
>
> We adapted the references in the revised manuscript making it more clear to which figure/table/equation we refer.
>
> In summary, these revisions and additions address your concerns by enhancing the objective evaluation of our approach and justifying our methodological choices. We believe these changes strengthen the manuscript and look forward to your further feedback.

---

> > ### Author Response · Authors · 2023-11-19
> > **References for Answer to Reviewer**
> >
> > [1]      	I. Higgins et al., “β-VAE: LEARNING BASIC VISUAL CONCEPTS WITH A CONSTRAINED VARIATIONAL FRAMEWORK,” 2017.
> > [2]      	G. W. Taylor, G. E. Hinton, and S. Roweis, “Modeling Human Motion Using Binary Latent Variables”.
> > [3]      	copper.hat, “Answer to ‘How to prove a set of positive semi definite matrices forms a convex set?,’” Mathematics Stack Exchange. Accessed: Nov. 18, 2023. [Online]. Available: https://math.stackexchange.com/a/322743
> > [4]      	A. Feragen and A. Fuster, “Geometries and Interpolations for Symmetric Positive Definite Matrices,” in Modeling, Analysis, and Visualization of Anisotropy, T. Schultz, E. Özarslan, and I. Hotz, Eds., in Mathematics and Visualization. , Cham: Springer International Publishing, 2017, pp. 85–113. doi: 10.1007/978-3-319-61358-1_5.

---

> ### Comment · Reviewer_PFFz · 2023-11-22
>
> Thanks for these answers.
>
> I still have some concerns of the lack of evaluation with respect to other possible approaches. It is especially difficult to see the benefit of the proposed quantitative metric without further details. For instance, if to consider work done in [1] how could we position it here. In [1] use of auto-encoder is also proposed as well as a quality metric based on prediction metrics on further patterns. When I was suggestion 3rd pattern this was to go into that direction. To see how the proposed generation model could predict intermediate patterns.
>
> Considering feedback on $\beta$ parameters also provide a mixed feeling since finally it seems quite sensitive on the considered use case as said in response from authors.
>
> So i am rather enclined to keep my initial rating.
>
> [1] Hughet et al, 'Manifold interpolating Optimal-Transport flows for trajectory inference', Neurips 22.

---

> ### Author Response · Authors · 2023-11-23
> **Reply to comment**
>
> Thank you for your constructive feedback, which has led to significant improvements in our manuscript.
>
> **Interpolating trajectories is more difficult than filling out single-time steps**
>
> We thank you for your feedback and reference to the “MIOFlow” framework by Huguet et al.
> However, we would like to note that while both CARAE and MIOFlow leverage auto-encoder concepts to analyze time series, their aim, methodology, and applications diverge significantly.
> - **Very different task**: CARAE focuses on interpolating between trajectories conditioned on a latent variable. MIOFlow only deals with filling held-out time points within a trajectory. They never use their methods on novel interpolated trajectories.
> - **Evaluation**: because they don’t evaluate on interpolation between trajectories, it’s much easier to quantify by holding out a one-time point. We also remark that the referenced paper relies on many qualitative comparisons.
> - **Different degree of difficulty**: We would like to argue that their task is relatively simpler because:
> Their task does not involve long-term stability issues because they deal with very short trajectories with simple dynamics: max 5-time steps versus around 150-time steps (of dimension around 100) for our work. Also, the video in the supplementary material shows that our pattern generator is very stable through transitions and at various levels.
> Our approach emphasizes the controlled generation of continuous pattern families from extremely limited data (as such data might be difficult to generate, particularly in RL settings, see other comments).
> They do  not propose a mechanism to morph between trajectories while they are unfolding (as we showed in section 4.3).
> - **The bottleneck definition**:  MIOFlow employs a combination of dynamic models, manifold learning, and optimal transport to interpolate single timesteps between static population snapshots.
> While MIOFlow emphasizes a probabilistic framework to define their bottleneck, CARAE emphasizes the geometry of neural dynamics. MIOFlow's use of Neural ODEs and its emphasis on optimal transport with manifold ground distance distinguishes it from CARAE, which prioritizes the linear interpolation of conceptors and the generation of patterns specifically within a defined probabilistic framework.
>
> **Benchmarking with a quantitative measure.**
>
> We have now addressed your suggestion for quantitative evaluation using a third pattern to benchmark the performance of our CARAE framework, taking inspiration from the MIOFlow paper that you added as reference.
>
> In our revised approach, detailed in Appendix I, we compared the performance of CARAE against a baseline method of simple linear interpolation between two distinct temporal motion patterns: walking and running. Our goal was to generate an intermediate time series that would ideally represent a jogging motion (the “correct” intermediate between walking and running from the dataset), and then measure the accuracy of these generated patterns against the ground truth jogging data available in our dataset.
>
> We used MSE and JSTCC the as the metric for quantification. The results, as presented in Appendix I, demonstrate that CARAE achieves a lower MSE in comparison to the direct linear interpolation approach when evaluated against the ground truth jogging data. This finding substantiates the efficacy of CARAE in generating more accurate intermediate temporal patterns and also provides a clear, objective measure of its performance. This establishes a tangible benchmark for future research, offering a solid foundation upon which to evaluate, measure, and enhance our proposed framework, task, and solution.
>
> We believe that this additional experiment and its results effectively address your concerns regarding the lack of quantitative evaluation and comparative analysis in our initial submission. The ability of CARAE to outperform a straightforward linear interpolation method in this context highlights its potential as a more nuanced and effective tool for interpolating between temporal motion patterns.
>
> We hope that this new evidence, alongside the other revisions made in response to your comments, strengthens the case for the novelty and utility of our proposed CARAE framework.

---

### Official Review · Reviewer_cFnM · 2023-10-31

**Soundness:** 2 fair
**Presentation:** 3 good
**Contribution:** 3 good
**Rating:** 6
**Confidence:** 1

**Summary:**

The authors introduce a novel algorithm called "Conceptor-Aided Recurrent Autoencoder" (CARAE) to address the challenge that generalizing from just two time series to create intermediate patterns in the context of representation learning. In addition, the author show the effectiveness proposed method on mocap motion modeling .

**Strengths:**

- The method section is well prepared and concise.

**Weaknesses:**

In all honesty, I'm not well-versed in this particular subject, and I've noticed that there are limited related works available for reference. Despite dedicating a significant amount of time to study the topic, I'm still finding it challenging to provide a comprehensive and professional feedback on the paper.


- Limited Related Works: Research can be more challenging when there are few related works to use as a basis for comparison or to gain a deeper understanding of the context.

- Outdated References: The reference to Jaeger (2014) being eight years old may indicate that the paper doesn't incorporate more recent developments in the field. It's common in rapidly evolving fields like machine learning for research to become outdated relatively quickly.

**Questions:**

1. Why using RNN rather than more recent and powerful transformer architecture?
2. Why there is not a comparison with related works or baseline?
3. It's hard to get that (b) in Figure 4 is intermediate pattern.

---

> ### Author Response · Authors · 2023-11-19
> **Answer to Reviewer**
>
> We thank you for your feedback, which we would like to address as follows
>
> 1. Acknowledging the Relevance of Older References:
> We would like to point out that the age of a reference does not undermine its validity. In fact, older concepts often find renewed significance in contemporary research. Additionally, linear subspace characterization linear subspace control did have recent successes by being effectively combined with backpropagation to address catastrophic forgetting issues, as highlighted in recent studies [1], [2]. Also, the concept of low-dimensional dynamics has recently gained traction in computational neuroscience, serving as a foundation for rich theoretical and experimental work to understand the dynamical aspect of the brain [3]. Our research proposes a novel interpretation of these phenomena by connecting it to the rich field of learning representation and autoencoding in Deep Learning.
>
> 2. Differentiating from Jaeger’s Work and Aligning with Current SOTA Research:
> Our approach represents a significant departure from Jaeger’s initial work [4], where linear subspace control was not linked to learning with RNNs but to store memories akin to Hopfield Networks but for time series. Instead, our methodology is more closely aligned with contemporary SOTA research in meta-learning [5], where some meta-parameters (equivalent of the conceptor) are multiplicatively interacting with a base model to improve the learning data-efficiency. These recent studies recognize and utilize the potential of linear subspace control in developing advanced learning models.
>
> 3. Emphasizing the Role of RNNs in Deep Learning:
> In the realm of deep learning, it's crucial to note that RNNs stand on par with other architectures like transformers on data-sparse domains. This includes the distillation and compression of locomotion policies into multi-task policy networks [6, 7]. Finding good compression principles is recognized as promising to be more data-efficient. Moreover, various RNN architectures are increasingly challenging transformers in terms of reducing computational costs and offering unbounded context length [7]. Additionally, some RNNs architectures have been shown to be very close to attention-based computations  [8].
>
> 4. Extending the CARAE Architecture:
> As mentioned in our responses to other reviewers, we have expanded our work in the appendix. Here, we demonstrate that the CARAE (Conceptor-Aided Recurrent AutoEncoder) architecture can be extended to scenarios involving more data, higher-dimensional bottlenecks, and other commonly used architectures like the LSTM. Furthermore, we adapted the CARAE to static Feed Forward Networks (FFWs) like MLPs applied to time series prediction, theoretically allowing for application to transformers.
>
> [1]      	X. He and H. Jaeger, “Overcoming Catastrophic Interference using Conceptor-Aided Backpropagation,” presented at the International Conference on Learning Representations, Feb. 2018. Accessed: Aug. 21, 2023. [Online]. Available: https://openreview.net/forum?id=B1al7jg0b
> [2]      	L. Duncker, L. Driscoll, K. V. Shenoy, M. Sahani, and D. Sussillo, “Organizing recurrent network dynamics by task-computation to enable continual learning,” in Advances in Neural Information Processing Systems, Curran Associates, Inc., 2020, pp. 14387–14397. Accessed: Aug. 17, 2023. [Online]. Available: https://proceedings.neurips.cc/paper/2020/hash/a576eafbce762079f7d1f77fca1c5cc2-Abstract.html
> [3]      	“Low-rank RNNs in ten minutes,” neurosopher. Accessed: May 31, 2023. [Online]. Available: https://adrian-valente.github.io/2022/06/01/low-rank-summary.html
> [4]      	H. Jaeger, “Controlling Recurrent Neural Networks by Conceptors,” arXiv:1403.3369 [cs], Apr. 2017, Accessed: Dec. 09, 2020. [Online]. Available: http://arxiv.org/abs/1403.3369
> [5]      	N. Zucchet, S. Schug, J. von Oswald, D. Zhao, and J. Sacramento, “A contrastive rule for meta-learning.” arXiv, Oct. 03, 2022. Accessed: Jul. 07, 2023. [Online]. Available: http://arxiv.org/abs/2104.01677
> [6]      	J. Merel et al., “Neural probabilistic motor primitives for humanoid control.” arXiv, Jan. 15, 2019. Accessed: May 05, 2023. [Online]. Available: http://arxiv.org/abs/1811.11711
> [7]      	N. Wagener, A. Kolobov, F. V. Frujeri, R. Loynd, C.-A. Cheng, and M. Hausknecht, “MoCapAct: A Multi-Task Dataset for Simulated Humanoid Control.” arXiv, Jan. 13, 2023. doi: 10.48550/arXiv.2208.07363.
> [8]      	D. Y. Fu, T. Dao, K. K. Saab, A. W. Thomas, A. Rudra, and C. Ré, “Hungry Hungry Hippos: Towards Language Modeling with State Space Models.” arXiv, Apr. 28, 2023. doi: 10.48550/arXiv.2212.14052.
> [9]      	“RNNs strike back,” neurosopher. Accessed: Nov. 18, 2023. [Online]. Available: https://adrian-valente.github.io/2023/10/03/linear-rnns.html

---

### Official Review · Reviewer_T2vx · 2023-11-01

**Soundness:** 3 good
**Presentation:** 3 good
**Contribution:** 3 good
**Rating:** 6
**Confidence:** 2

**Summary:**

The paper proposes to use conceptors (soft-projection matrices) to constrain the dynamics of an RNN to a low dimensional geometry space when learning temporal patterns in a two-shot regime. By encouraging the conceptors corresponding to the two input patterns to be close to each other when training in an autoencoder setup, the RNN learns a manifold of conceptors that can be traversed to generate interpolations, similar to the latent space traversal in a VAE.
The proposed method is validated with a simple sine waves example and with an example of interpolating between walking and running using 2 sequences from the Mocap dataset.

**Strengths:**

The idea of learning to control the dynamics of an RNN with only two-shots is very appealing.
The proposed formulation is simple and appears to work very well for the given examples.
The method seems to be novel, but I do not have enough familiarity with the topic to give a stronger assessment.
The paper is well written.

**Weaknesses:**

If the two sequences differ in multiple underlying factors, would the learnt interpolation still give sensible results? E.g. given 2 moving mnist sequences, that differ in the colour of the digits (red vs blue on black background) and the digits themselves (e.g. 0s vs 8s), what would the interpolations look like?

Were the values of \beta_1 and \beta_2 (in eq 8) ablated?

Small comment: in Fig 3 (c->j), it is difficult to distinguish the different colours.

Post rebuttal and after checking the other reviews, I decided to reduce the score to 6. Indeed, the evaluation could be improved.

**Questions:**

See above.

---

> ### Author Response · Authors · 2023-11-19
> **Answer to Reviewer**
>
> Thank you for your valuable feedback. Regarding the interpolation of different features, we have not yet incorporated a way to address different features independently. That means, in terms of your proposed colored digit interpolation where only 2 examples are given, both features (digit pattern and color) are interpolated at the same time guided by interpolating the conceptor during inference. However, and this would be partially related to the answer to reviewer 1, learning these disentangled factors of variation (digit pattern and colors) would require extending CARAE to the case where more than two-time series are present and a bottleneck of more than one dimension (in conceptor space). The multi-dimensional bottleneck is critical to discover the different factor of variations and to allow for feature-specific interpolation. Furthermore, once CARAE is expanded in this manner, it becomes possible to adapt techniques from the auto-encoder literature to promote disentanglement. For instance, one can enforce a probabilistic interpretation of the bottleneck and ensure the independence of its dimensions, as seen in VAE and Beta-VAE methods.
>
> It's worth mentioning that the disentanglement of factors of variation holds significant relevance in the context of motion modeling. This is particularly important because a key challenge in achieving realism in simulating characters is the ability to portray various behaviors with distinct styles, such as depicting characters as drunk, sad, or excited [1].
>
> In the appendix, we introduced a similarity metric (see the updated manuscript) to assess the connection between the intermediary inferences of the CARAE framework and the probabilistic attributes and time-related correlations to the training datasets. Utilizing this metric, we conducted an ablation analysis on the regularization parameters beta_1 and beta_2.
>
> [1] Taylor, Graham & Hinton, Geoffrey. (2009). Factored conditional restricted Boltzmann Machines for modeling motion style. Proceedings of the 26th International Conference On Machine Learning, ICML 2009. 129. 10.1145/1553374.1553505.

---

### Official Review · Reviewer_Wnkt · 2023-11-01

**Soundness:** 2 fair
**Presentation:** 3 good
**Contribution:** 1 poor
**Rating:** 3
**Confidence:** 3

**Summary:**

The work proposed CARAE an approach of learning a continuous spectrum of temporal representation from two sequence of training examples. The proposed approach is based on the existing study of Conceptor and a matrix conceptor $C$ is inserted into an RNN model to control the update dynamics of the RNN's internal states. Two separate $C$s are learned from the two training sequences and a regularization term in the training objective attempts to minimize their distance. Interpolation is performed between the two $C$s to generate sequences that have temporal patterns interpolating between the two training sequences.

**Strengths:**

The proposed problem setting is indeed challenging and novel and despite the challenging setting. Qualitative results from experiments on both synthetic data and real-world MOCAP data suggest the capability of the proposed CARAE in learning interpolatable representation from two training sequences.

**Weaknesses:**

The major weakness of the work is a lack of justification for its practical significance, which makes it difficult to judge the work's contribution.
1. It is hard for me to judge the practical significance of the problem settings of the work, learning a continuous spectrum of representations that can interpolate between two training examples. The work mentioned related works of interpolatable representation learning in different areas, including locomotion modelling, robotics, and reinforcement learning. None of the mentioned work studies problems settings even close to the proposed setting. The proposed setting also restricts the approach to a two-shot learning setting and it is not clear how the approach can be extended to few-shot learning settings with more training examples and if the proposed approach would still have any practical value when larger amount of training data is available, which is not unusual in practice.
2. The proposed CARAE is based on an artificial RNN model with specific architecture designs, which is different from more commonly used RNN architectures like LSTM or GRU. Extending CARAE to these more common RNN architectures or more recent transformer architecture could significantly improve its practical value.

Apart from my concerns on practical significance, the work also have the following minor weakness:
1. In Sec. 5, the work uses *two data sets* and *two distinct examples* to call $u_1$ and $u_2$ which can be confusing. It would be much more clear if $u_1$ and $u_2$ are referred to as two training examples consistently in the work.
2. The work only presents qualitative results and training loss curves. The lack of a systematic quantitive evaluations makes the proposed approach and claimed contributions less convincing.

**Questions:**

In Sec. 2, the work identifies 4 challenges in the ability of RNN to generalize to different temporal dynamics. **Inferences** is explicated tackled with in Sec. 3. Does CARAE address the other three challenges? If so, how are they addressed?

---

> ### Author Response · Authors · 2023-11-19
> **Answer to Reviewer**
>
> We thank you for your valuable feedback on our work. We would like to address your concerns step-by-step in the following:
>
> 1. 1.  We mention related works of interpolatable representation learning in different areas, including locomotion modeling, robotics, and reinforcement learning because they deal with certain challenges, like sparsity of data while aiming for stable periodic solutions that can be interpolated. We would like to argue that the CARAE framework, although not yet applied to RL learning tasks, shows promising results in the motion modeling using the MoCap data set. Thereby, we address and overcome several of the discussed challenges that yet could only be solved by massive computational effort.
>
>     2.  While our current focus is on two-shot learning, we see the potential for expanding CARAE to few-shot learning scenarios. Thereby, the loss function needs to be reformulated to include several conceptors, one per learned temporal pattern. The CARAE framework even enables us to encode into the loss function whether the patterns should be all-to-all interpolated or whether they should line up along a single dimension. We briefly outline potential extensions of our method in the manuscript's conclusion and in appendix E, discuss how CARAE could adapt to and benefit from additional training examples, thereby enhancing its practicality and applicability in a broader range of data-constrained environments.
>
>
> 2. We appreciate your concern about extending CARAE to more commonly used RNN architectures like LSTM and to transformer architectures. As CARAE is an architecture-independent mechanism for manipulating neural dynamics through conceptors, we did not expect any problems with this. Indeed, in response to your suggestion, we have extended CARAE to the commonly used LSTM architecture. Furthermore, we have outlined a version of the CARAE that relies on a feed-forward neural net. Both architectures are able to learn continuous interpolation of temporal patterns as we present in the appendix C. These extensions demonstrate CARAE’s versatility and significantly broaden CARAE's practical applicability.
>
> Minor weaknesses:
>
> 1. We thank  you for the careful reading and have streamlined the formulation in section 5 to avoid any confusion.
>
>
> 2. In the appendix, we added the introduction of a similarity measure (see revised manuscript) that evaluates the relation of our inferences to probabilistic features and temporal correlations of the training datasets. Based on this metric, we performed an ablation study of the two regularization parameters beta_1 and beta_2. The metric thereby adds a quantitative measure to evaluate the inferences beyond the loss terms we design, allowing for a more systematic investigation of our proposed method.
>
> Regarding further questions:
>
> The CARAE framework effectively addresses all four challenges outlined in Section 2. The incorporation of the conceptor in the autoregressive mode ensures stable dynamics, protecting against exploding dynamics, interferences, and side dynamics. Our explicit training approach within the CARAE loss term thereby specifically tackles fixed-point dynamics, a critical aspect when aiming for interpolation. These improvements are needed to yield stable interpolated temporal patterns when dealing with sparse data as showcased by the two experiments discussed in the manuscript. Additionally to the failing vanilla RNN used in the sine wave interpolation, we added unsuccessful experiments using a RAE on both training sets to the appendix.
>
> In conclusion, we believe that our revisions and additional analyses respond comprehensively to your concerns, thereby significantly strengthening the contribution and clarity of our work. We look forward to further feedback!

---

> ### Comment · Reviewer_Wnkt · 2023-11-21
> **Post-rebuttal response**
>
> I would like to thank the authors for the detailed response. Maybe I'm not the target audience of this work but I'm not fully convinced by the authors' arguments for the practical significance of its unconventional problem setting. The response acknowledges the importance of learning interpolatable representation in many areas including reinforcement learning, robotics, and locomotion modelling. To my best knowledge, none of existing works in these areas that showcased interpolatable representation has a similar problem setting. Can the author give examples of existing works that adopt such a setting or examples of realistic problem settings where learning interpolatable representation from two samples is necessary? In addition, I'm not sure if the results on MoCap data are enough to justify the practical significance of the work. Controllable human motion generation under various settings has been studied in many existing works [1, 2]. Moreover, there's no unique way of interpolating between two human motion trajectories that looks realistic. Therefore I do not see a direct and objective way of evaluating the interpolated trajectories except for indirect and qualitative evaluations. I would encourage the author to spend more efforts on justifying the practical importance the problem setting for a more general audience and design experiment settings to help support that in future revision.
>
>
> [1] Petrovich, Mathis, Michael J. Black, and Gül Varol. "Action-conditioned 3D human motion synthesis with transformer VAE." Proceedings of the IEEE/CVF International Conference on Computer Vision. 2021.
>
> [2] Kania, Kacper, Marek Kowalski, and Tomasz Trzciński. "TrajeVAE: Controllable Human Motion Generation from Trajectories." arXiv preprint arXiv:2104.00351 (2021).

---

> > ### Author Response · Authors · 2023-11-22
> > **Answer to Reviewer part 1/2**
> >
> > We would like to thank you for engaging with us. However, we challenge the assumption that our novel representational learning algorithm lacks practical importance. Contrary to your references, our work focuses on the smooth interpolation between learned motions, not just generating motion patterns from sequences. This is not the main aim of the work we present here. Other works, like [1], focus on learning velocity control in human motion using reinforcement learning. However, these approaches, lacking a robust representational learning mechanism, often require extensive training and computation, often resulting in unnatural motion patterns. In contrast, our work introduces a method for learning continuously interpolatable abstract representations using a small RNN trained on sparse data. In the following, we will address the raised points more in detail:
> >
> > **First method that shows interpolatable generalization relying on two-shot learning**: We focussed on the two-shot learning setting to showcase the unique generalization capability. Thereby, we present that our framework outperforms other e.g. classical recurrent auto-encoders that are notoriously hard to train and require a lot of data. Thereby we are less constraining, as there is no need to encode all the input into a low-dim vector space as done e.g. in transformer-based architectures. Furthermore, in the last revision, we introduced how our setup is extendable to few shot learning which further extends the applicability of our method.
> >
> > **Relevance for applications in RL**: Two-shot and few-shot learning are central challenges in machine learning. For instance, the recent resurgence of meta-learning is motivated by the drastic difference between humans and machines in terms of sample efficiency (cf. omniglot [5], finn). In RL, this need is more critical, as current algorithms still lag behind human-level sample efficiency. Addressing the reviewer’s question about related works, we would highlight the work of Wayne et al[2], Mocapact[1], and more recently Gehring[2]. These start from a few policies learned by imitation (walking, running, jogging) and aim to distill them into a single neural network to accelerate further reinforcement learning on downstream tasks like following a speed target (cf. Figure). Despite showing minor improvements like faster subsequent learning and better pose, they still rely on significantly more training to follow intermediary speed targets. In our work, we accomplished a more complex task by successfully enabling the model to follow intermediary speed targets without requiring additional training (cf. video in Supplementary Materials), a development that shows a promising direction for reinforcement learning applications.

---

> > > ### Author Response · Authors · 2023-11-22
> > > **Answer to Reviewer part 2/2**
> > >
> > > **Qualitative judgement is important for motion-related tasks and generative models in general**: We agree with the reviewer that “there's no unique way of interpolating between two human motion trajectories that looks realistic.” However, there are many ways of interpolating that don’t look realistic. Here are some qualitative features that are still challenging to tackle in motion modeling
> > > - fixed point issues: when generating in an autoregressive mode, models tends to fall into a fixed pose [4]. This makes it hard to model periodic behavior.
> > > - transition issues: when transitioning between different motion behaviors, models display glitches [6]
> > >
> > > These are easy to spot by qualitative judgment but way more harder to quantify. Even the papers mentioned by the reviewer employ “plausible” and qualitative judging. Generative models are better judged with qualitative statements. Other generative models such as LLM are judged by qualitative measures as well as it is that hard to quantify e.g. “hallucination”.
> > >
> > > Although we recognize the importance of applied work, our CARAE framework proposes a novel principle for dynamic data representation learning. Thereby, we introduce a new prior to learn the representation based on the geometry of the neural dynamics which as indicated in the appendix of our work can be applied to various architectures. Our introduce method allows for visually convincing motion interpolation based on learning from only two data sets which showcase its strong ability to learn and abstract representations.
> > >
> > > [1] Wagener, Nolan, et al. "MoCapAct: A Multi-Task Dataset for Simulated Humanoid Control." Advances in Neural Information Processing Systems 35 (2022): 35418-35431.
> > >
> > > [2] Merel, Josh, et al. "Learning human behaviors from motion capture by adversarial imitation." arXiv preprint arXiv:1707.02201 (2017).
> > >
> > > [3] Gehring, Jonas, et al. "Leveraging Demonstrations with Latent Space Priors." arXiv preprint arXiv:2210.14685 (2022).
> > >
> > > [4] Z. Ye, H. Wu, and J. Jia, “Human motion modeling with deep learning: A survey,” AI Open, vol. 3, pp. 35–39, Jan. 2022, doi: 10.1016/j.aiopen.2021.12.002.
> > >
> > > [5] Lake, B. M., Salakhutdinov, R., and Tenenbaum, J. B. (2015). Human-level concept learning through probabilistic program induction. Science, 350(6266), 1332-1338.
> > >
> > > [6] J. Merel et al., “Hierarchical visuomotor control of humanoids.” arXiv, Jan. 15, 2019. Accessed: Aug. 10, 2023. [Online]. Available: http://arxiv.org/abs/1811.09656

---

### Meta-Review · Area_Chair_w9yJ · 2023-12-10

**Metareview:**

This work proposed CARAE to use recurrent auto-encoder with a conceptor-based regularization to generate sequences that have temporal patterns interpolating between the two training sequences. The reviewers think that the problem that this paper tries to tackle is challenging, but have concerns on lacking of practical significance, which is not resolved in the rebuttal period. There is also unresolved concerns on lacking evaluation and comparison with other approaches.

Hence, rejection is recommended for this paper. The authors are encouraged to improve the quality of the draft based on the reviewers feedback.

**Justification For Why Not Higher Score:**

motivation justification and sufficient evaluation

**Justification For Why Not Lower Score:**

N/A

---

### Decision · Program_Chairs · 2024-01-16

Reject